# HiPPO Zoo: Explicit Memory Mechanisms for Interpretable State Space Models

Jack Goffinet [1]   Casey Hanks [1]   David E. Carlson [1 2]

## Abstract

Representing the past in a compressed, efficient, and informative manner is a central problem for systems trained on sequential data. The *HiPPO* framework, originally proposed by Gu & Dao et al., provides a principled approach to sequential compression by projecting signals onto orthogonal polynomial (OP) bases via structured linear ordinary differential equations. Subsequent works have embedded these dynamics in state space models (SSMs), where HiPPO structure serves as an initialization. Nonlinear successors of these SSM methods such as Mamba are state-of-the-art for many tasks with long-range dependencies, but the mechanisms by which they represent and prioritize history remain largely implicit. In this work, we revisit the HiPPO framework with the goal of making these mechanisms explicit. We show how polynomial representations of history can be extended to support capabilities of modern SSMs such as adaptive memory allocation and associative memory, while retaining direct interpretability in the OP basis. We introduce a unified framework comprising five such extensions, which we collectively refer to as a "HiPPO zoo." Each extension exposes a specific modeling capability through an explicit, interpretable modification of the HiPPO framework. The resulting models adapt their memory online and train in streaming settings with efficient updates. We illustrate the behaviors and modeling advantages of these extensions through a range of synthetic sequence modeling tasks, demonstrating that capabilities typically associated with modern SSMs can be realized through explicit, interpretable polynomial memory structures.

## 1. Introduction

Learning effective representations of past inputs under fixed memory constraints is a central challenge in sequence modeling, requiring models to maintain a compressed representation of history that supports accurate and flexible processing of long sequences, including the ability to handle long-range dependencies and input-dependent state updates. One increasingly successful approach is the state space model (SSM) paradigm,[1] which maintains a fixed-dimensional state vector that is updated autoregressively by incoming inputs, $\mathbf{s}_{t+1} = f(\mathbf{s}_t, y_t)$. Compared to context-window-based approaches, SSMs scale efficiently to long sequences and achieve strong empirical performance across a variety of sequence modeling tasks. However, in modern SSMs, the mechanisms by which history is represented, prioritized, and transformed are largely implicit, encoded in learned state dynamics that are difficult to interpret or analyze directly.

A key advance toward making memory representations explicit is the HiPPO (High-Order Polynomial Projection Operators) framework introduced by Gu et al. (2020), building on the earlier work of Voelker et al. (2019). In HiPPO, the past of a one-dimensional continuous-time signal is represented by its projection onto an orthogonal polynomial basis under a fixed weighting function (or, more generally, a measure). The resulting coefficient vector, $\mathbf{s}$, provides a direct and interpretable representation of the signal's history, and can be maintained online via a structured linear dynamical system. In contrast to modern SSMs, where the state vector is an opaque learned summary of the past, HiPPO exposes the contents of memory explicitly through its polynomial structure and associated weighting function.

Despite its explicit and interpretable representation of history, the original HiPPO framework does not incorporate several modeling capabilities that are central to modern state space models. Contemporary SSMs routinely employ input-dependent state updates, adaptive allocation of memory across timescales, nonlinear interactions between past and present, and mechanisms akin to associative memory (Gu et al., 2022b; Smith et al., 2023; Gu & Dao, 2024), all

---

[1]Department of Computer Science, Duke University, Durham NC, USA [2]Departments of Civil and Environmental Engineering; Biostatistics and Bioinformatics; and Electrical and Computer Engineering, Duke University, Durham NC, USA. Correspondence to: Jack Goffinet <jack.goffinet@duke.edu>.

*Proceedings of the 43$^{rd}$ International Conference on Machine Learning*, Seoul, South Korea. PMLR 306, 2026. Copyright 2026 by the author(s).

---

[1]Throughout this work, "state space model" refers to modern neural sequence models popularized by S4 and related work, not to classical probabilistic state space models emphasizing latent state estimation.

of which contribute to their empirical success on complex sequence modeling tasks. These capabilities are realized implicitly through learned state dynamics that sacrifice direct interpretability.

We revisit the HiPPO framework with the goal of extending it to support these modern capabilities while preserving its explicit, interpretable structure. We show that many capabilities introduced in modern SSMs can be realized within HiPPO as explicit modifications to the underlying history measure or the dynamics governing the polynomial coefficients. This perspective yields a unified framework in which these memory mechanisms are directly represented in an interpretable orthogonal polynomial basis, rather than encoded implicitly in learned state transitions. By making these mechanisms explicit, the models retain the structural and computational advantages of HiPPO while enabling principled analysis of how past information is stored and used.

We introduce a framework comprising five such extensions, which we collectively refer to as a "HiPPO zoo." [2] Volterra HiPPO incorporates nonlinear interactions by embedding Volterra series directly into the orthogonal polynomial basis. Salience HiPPO introduces input-dependent warping of the history measure, allowing the system to allocate memory preferentially to informative inputs. Associative Memory HiPPO embeds explicit associative structure into the polynomial representation, enabling interpretable key–value memory. Multiscale HiPPO represents memory across multiple timescales by explicitly embedding a continuum of HiPPO states. Finally, Forecasting HiPPO makes explicit how forecasting objectives shape memory, revealing how different horizons induce different predictive memories and geometries on the past.

Across a range of sequence modeling experiments, we use these extensions to illustrate how explicit polynomial memories can realize behaviors that are implicit in modern state space models. Our experiments probe how memory is allocated, transformed, and queried under different mechanisms, and visualize these effects directly through the orthogonal polynomial basis. Together, these results show that making memory structure explicit enables principled analysis and transparent design of sequence models, while retaining the structured linear dynamics and efficient online updates of the HiPPO framework.

## 2. Background and Related Work

**Orthogonal Polynomials** Orthogonal polynomials (OPs) provide a convenient and interpretable basis for representing functions with respect to a chosen weighting over their

domain. Given a weighting function $\omega(x)$ on a domain $\mathcal{X} \subseteq \mathbb{R}$ (or more generally a positive measure), an orthonormal polynomial sequence $\{P_n(x)\}_{n=0}^{\infty}$ forms a basis under the inner product $\langle P_m, P_n \rangle_\omega = \delta_{mn}$, allowing a function to be represented in terms of coefficients that depend explicitly on $\omega$. The choice of $\omega$ plays a central role: it defines which regions of the past are represented most accurately, and thus how memory is allocated over time. A uniform weighting, for instance, weights all positions within a finite window equally while an exponential weighting discounts the distant past continuously, corresponding to the Leg-T and Leg-S systems used throughout this paper. Throughout this work, we use OP bases to obtain compact and structured representations of sequence history, where each coefficient corresponds to a well-defined linear functional of the past. For completeness, additional background on OPs and their properties is in Appendix A.

**HiPPO** The HiPPO (high-order polynomial projection operators) framework provides a principled method for maintaining an explicit representation of the recent past of a signal in an online setting (Voelker et al., 2019; Gu et al., 2020). At each time $t$, the value of a one-dimensional input function $f$ at a lag $\tau \geq 0$ is approximated by a truncated orthogonal polynomial expansion $\sum_{n=0}^{N-1} s_n(t) P_n(\tau)$, where the basis $\{P_n\}_{n=0}^{N-1}$ is defined with respect to a weighting function $\omega(\tau)$ that specifies the relative importance of accurately representing different regions of the past. The coefficient vector $\mathbf{s}(t)$ therefore constitutes an explicit, interpretable representation of the history of the signal, with each coefficient corresponding to a well-defined linear functional of the past.

A key property of HiPPO is that these coefficients can be maintained online via a structured linear dynamical system

$$\dot{\mathbf{s}}(t) = A\mathbf{s}(t) + \mathbf{b}f(t) , \qquad (1)$$

where the dynamics matrix $A$ and input vector $\mathbf{b}$ are determined by the choice of polynomial basis and weighting function. Different choices of $\omega$ give rise to different HiPPO schemes, corresponding to distinct memory profiles over the past. In this work, we focus on the translated and scaled Legendre HiPPO variants (Leg-T and Leg-S, see Appendix B), which have been shown to provide stable and well-conditioned polynomial representations of history (Voelker et al., 2019; Gu et al., 2020; 2023). The appeal of HiPPO lies in its combination of an interpretable memory representation with structured linear dynamics that admit efficient online updates, making it a natural substrate for exposing and analyzing memory mechanisms in state space models. Owing to its continuous-time formulation, HiPPO also naturally accommodates irregular sampling and varying sampling rates (e.g., Graf et al., 2025), though these aspects are not the focus of this work.

**Neural State Space Models (SSMs)** Early work on neural state space models embeds continuous-time dynamical systems of the form in Eq. (1) within recurrent neural networks, often augmented with nonlinearities analogous to those used in LSTMs or GRUs (Voelker et al., 2019; Gu et al., 2020). In this setting, the state vector evolves according to linear dynamics driven by the input signal, while nonlinearities are introduced either within or around the state update to increase expressivity. Gu et al. (2021) observed that many of these nonlinearities could be moved outside the state evolution itself, leading to a class of linear state space models coupled with learned input and output projections that achieve strong empirical performance on a range of sequence modeling tasks.

Linear SSMs typically maintain a continuous- or discrete-time state vector governed by linear dynamics, together with a learned readout of the form $y(t) = W\mathbf{s}(t)$, where $W$ maps the internal state to task-specific outputs. A series of works has refined this basic formulation to improve numerical stability, efficiency, and scalability. The S4 model (Gu et al., 2022b) introduced techniques for representing the linear recurrence induced by Eq. (1) as a convolution, enabling efficient parallel training. Subsequent models such as S5 (Smith et al., 2023) extended this approach to multi-input–multi-output settings using fast parallel scan algorithms, while other works explored multiscale architectures (Goel et al., 2022) and structured product kernels for images and video (Nguyen et al., 2022). Additional lines of work investigated diagonal dynamics matrices to further improve computational efficiency and robustness (Gupta et al., 2022; Gu et al., 2022a; Yu et al., 2024; Rusch & Rus, 2025).

Recently, state space models have been proposed as alternatives to attention-based architectures in large language modeling and other long-context settings (Dao et al., 2023; Gu & Dao, 2024; Dao & Gu, 2024). Unlike standard attention, which relies on pairwise token comparisons and typically scales quadratically with context length, SSMs scale linearly in sequence length and can therefore support much longer effective contexts under fixed computational budgets. Architectures such as H3 demonstrate that SSMs can perform simple associative recall tasks, while subsequent models including Mamba and Mamba-2 introduce input-dependent state updates and selective mechanisms that prioritize and retrieve past information conditioned on the current input (Dao et al., 2023; Gu & Dao, 2024; Dao & Gu, 2024). In these models, however, associative or content-addressable behavior is mediated through selective state updates rather than explicit key–value representations, making the underlying memory structure difficult to isolate or interpret.

Across these developments, modern SSMs have steadily increased their expressive power by incorporating input-dependent updates, multiscale dynamics, nonlinear interactions, and implicit associative behavior. However, these capabilities are typically realized through learned state transitions and parameterized update rules, making it difficult to directly interpret how different aspects of the past are stored, prioritized, or transformed within the state. In contrast to the explicit polynomial memory representations provided by HiPPO, the internal states of neural SSMs generally function as opaque summaries of history. The goal of this work is to demonstrate that such capabilities can be achieved within the explicit, interpretable polynomial framework of HiPPO, offering transparent alternatives for applications where mechanistic understanding is valued alongside predictive performance.

## 3. Methods and Results

The following sections introduce each member of the HiPPO zoo with experiments on synthetic data designed to highlight their explicit polynomial properties. We refer the reader to Appendix C for full experimental details, including complete descriptions of the data, models, and training procedures omitted from the main text.

### 3.1. Volterra HiPPO

Modern state space models rely on nonlinear input–output maps to implement nonlinear causal operators, typically by applying learned nonlinear functions (e.g., MLPs or gating mechanisms) to the state vector (Dao et al., 2023; Gu & Dao, 2024). A direct extension of HiPPO follows the same recipe: apply a nonlinear readout to the HiPPO state $\mathbf{s}(t)$ to produce the output. While effective in practice, this approach entangles nonlinear effects and obscures how specific interactions among past inputs contribute to the present. We instead pursue a structured alternative based on Volterra series, which decompose nonlinear dynamics into interpretable kernels of increasing order.

**Volterra Kernels in OP Bases** Volterra series provide a classical representation of causal, time-invariant nonlinear systems as a sum of multilinear functionals of the input history. Specifically, an input signal $f(t)$ is mapped to an output $y(t)$ through a collection of kernels $\{h_k\}_{k=0}^{\infty}$, where each $h_k$ captures the kth-order interactions:

$$y(t) = \beta^{(0)} + \sum_{k=1}^{\infty} \int_{(0,\infty)^k} h_k(\tau_1, \ldots, \tau_k) \prod_{j=1}^{k} f(t-\tau_j) \, \mathrm{d}\tau_j.$$

We observe that these kernels can be naturally parameterized in orthogonal polynomial bases, aligning directly with the structure of HiPPO memory.

Let $\{P_n\}$ denote OPs with respect to a weighting function

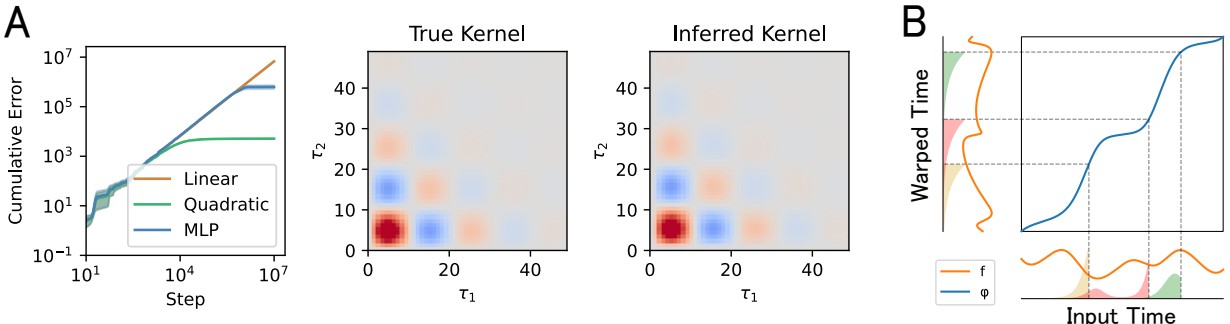

*Figure 1.* **A:** Online learning of a continuous time-invariant system. The ground truth system has a nonzero second-order Volterra kernel (middle). Online learning of a linear HiPPO system is insufficient to learn the kernel. Using a second-order (quadratic) Volterra HiPPO system, the correct kernel can be inferred (right), with faster convergence than an unstructured MLP readout (left, mean $\pm$ SEM across five independent runs). **B:** The time warping interpretation of Salience HiPPO. Salience HiPPO is equivalent to a standard HiPPO system under an invertible time warp $\varphi$. Consequently, the static weighting functions in warped time (left; green, red, and yellow) correspond to dynamic weighting functions in real time (bottom; green, red, and yellow).

$\omega(\tau)$. We represent the first few Volterra kernels as

$$h_0 = \beta^{(0)}, \quad h_1(\tau_1) = \omega(\tau_1) \sum_{i_1=0}^{\infty} \beta_{i_1}^{(1)} P_{i_1}(\tau_1) \,,$$

$$h_2(\tau_1, \tau_2) = \omega(\tau_1)\omega(\tau_2) \sum_{i_1,i_2=0}^{\infty} \beta_{i_1,i_2}^{(2)} P_{i_1}(\tau_1) P_{i_2}(\tau_2) \,,$$

and similarly for higher orders. Truncating the polynomial basis to degree $N$ yields coefficient tensors $\beta^{(k)} \in \mathbb{R}^{N^k}$, which explicitly parameterize interactions among lagged components of the input.

**Volterra HiPPO Readout** We use a HiPPO system to maintain an OP approximation of the recent past, $f(t-\tau) \approx \sum_{n=0}^{N-1} s_n(t) P_n(\tau)$, where $\mathbf{s}(t)$ is the HiPPO state vector. Substituting this expansion into the truncated Volterra series yields a polynomial readout of the form

$$y(t) = \beta^{(0)} + \sum_{k=1}^{K} \sum_{i_1,\dots,i_k=0}^{N-1} \beta_{i_1,\dots,i_k}^{(k)} s_{i_1}(t) \dots s_{i_k}(t) \,,$$

where $\sum_{i_1,\dots,i_k=0}^{N-1}$ denotes a sum over all order-$k$ ordered tuples of state vector indices, $i_1,\dots,i_k$, with each $i \in \{0,\dots,N-1\}$. In this formulation, memory is maintained by a linear dynamical system, while nonlinearity appears only in the readout. Each coefficient tensor $\beta^{(k)}$ directly encodes the strength of kth-order interactions among components of the past, providing a transparent alternative to black-box nonlinear readouts.

**Quadratic Volterra Example** For a second-order system (i.e., $h_k = 0$ for $k > 2$), the readout reduces to

$$y(t) = \beta^{(0)} + \mathbf{s}^{\top}\beta^{(1)} + \mathbf{s}^{\top}\beta^{(2)}\mathbf{s} \,,$$

yielding an explicit separation between linear and quadratic contributions of the past. We evaluate Volterra HiPPO on the nonlinear benchmark system of Wray & Green (1994), which admits a purely second-order Volterra representation. The parameters $\{\beta^{(k)}\}$ are learned online via gradient descent. As shown in Figure 1A, a linear Volterra HiPPO ($\beta^{(2)} = 0$) fails to capture the system dynamics, whereas a quadratic Volterra HiPPO accurately models the system,

revealing an interpretable interaction among pairs of lagged times. A single-hidden-layer MLP applied to the HiPPO state also learns the task, but converges more slowly and does not expose the underlying kernel structure, highlighting the interpretability advantages of the Volterra formulation.

## 3.2. Salience HiPPO

A central capability of modern state space models is their ability to adapt how past inputs are weighted in memory through learned, state-dependent update mechanisms. In selective SSMs such as Mamba and related architectures, this adaptivity is achieved through input-dependent state updates embedded in deep, residual dynamical systems. While these mechanisms are highly expressive, they do not expose how memory is allocated in an interpretable form.

*Salience HiPPO* makes this form of adaptive memory weighting explicit by introducing a scalar, time-dependent deformation of the HiPPO history measure. Rather than modifying the state update through high-dimensional gating, Salience HiPPO parameterizes adaptive memory weighting through a scalar, input-dependent deformation of the history measure. Informative inputs expand their local representation in the history measure, while uninformative inputs are compressed, yielding a transparent and analyzable mechanism for adaptive memory allocation.

Concretely, given an input signal $f(t)$ and a positive scalar salience signal $g(t)$, Salience HiPPO modifies Eq. (1) as

$$\dot{\mathbf{s}}(t) = g(t)\left[A\mathbf{s}(t) + \mathbf{b}f(t)\right] \,, \tag{2}$$

where the dynamics are scaled multiplicatively by $g(t)$. Larger salience increases the rate at which new information is incorporated into the state, explicitly prioritizing those timepoints in memory.

**Time Warping Interpretation** A useful perspective on Salience HiPPO is obtained via a change of variables. Define a warped time coordinate

$$t_1 = \varphi(t_0) \triangleq \int_0^{t_0} g(s) \mathrm{d}s , \tag{3}$$

where $t_0 = t$ is real time. Since $g(s) > 0$, this mapping is invertible. Under this transformation, (2) reduces to standard, time-invariant dynamics in the warped time coordinate,

$$\frac{\mathrm{d}\boldsymbol{s}(t_1)}{\mathrm{d}t_1} = A\boldsymbol{s}(t_1) + \boldsymbol{b}\tilde{f}(t_1) , \tag{4}$$

where $\tilde{f}(t_1) \triangleq f(\varphi^{-1}(t_1))$. Thus, Salience HiPPO is equivalent to applying a conventional HiPPO system to a reparameterized time axis, with the salience signal controlling the local rate of time.

**Induced History Measures** This perspective makes the effect of salience on memory explicit. Let $T$ denote the current real time, $s \le T$ a past input time, and $\omega_1(\tau_1)$ the standard HiPPO history measure over warped-time lag. The warped-time lag between $s$ and $T$ is $\tau_1(T, s) = \varphi(T) - \varphi(s) = \int_s^T g(u) \, \mathrm{d}u$, so the corresponding real-time history measure is

$$\omega_{0,T}(s) = \omega_1(\varphi(T) - \varphi(s)) \, g(s), \qquad s \le T. \tag{5}$$

Consequently, salience directly reshapes the history measure over real time: the weight assigned to a past input at time $s$ depends both on its local salience $g(s)$ and on its accumulated warped-time distance from the present. Because the OP basis is defined with respect to this measure, salience also alters the effective basis functions used to represent the past, yielding an adaptive representation of history (see schematic in Fig. 1B). This suggests a natural test: can a learned salience signal recover structured, interpretable memory allocation on a task that requires distinguishing informative from uninformative inputs?

**Selective Copying Experiment** We evaluate Salience HiPPO on an iterative selective copying task designed to probe adaptive memory allocation. Each episode consists of an input phase containing informative tokens interleaved with uninformative tokens, followed by a write phase where the model must output the informative tokens in order.

The model maintains a multi-channel Leg-T HiPPO memory and is trained online using truncated backpropagation through time. The salience signal $g(t)$ is produced by a small learned network that conditions on the current input token and a pooled summary of the current HiPPO state. All neural parameters are optimized online, while the HiPPO dynamics parameters $(A, \mathbf{b})$ are fixed.

As shown in the example episode in Figure 2, salience remains low during uninformative inputs and increases sharply for informative tokens. During the write phase, the model produces a faithful copy of the informative subsequence, despite having observed it interleaved with distractors. To understand the internal mechanism, we visualize the induced history measures at several timepoints. The measures allocate substantial weight to informative segments of the past while sharply down-weighting intervals corresponding to uninformative inputs. We further visualize the effective continuous-time linear output functionals during the write phase, which exhibit pronounced peaks precisely at the locations of the required input tokens. Together, these diagnostics show that Salience HiPPO solves the task by explicitly reshaping its history measure, making visible a form of input-dependent memory allocation that is typically implicit in modern state space models.

We emphasize that similar selective copying behavior can be achieved by selective SSMs such as Mamba, where input-dependent state updates implicitly modulate the effective integration timescale. The contribution of Salience HiPPO is not the adaptive mechanism itself, but its expression as an explicit deformation of the history measure within the HiPPO framework, enabling direct visualization and analysis of memory allocation in continuous time.

The selective-copying task isolates the ability to preserve sparse, salient inputs over long delays. As shown in Table 1, Salience HiPPO achieves 100.0% test accuracy, demonstrating that the scalar salience signal $g(t)$ successfully implements adaptive memory allocation despite being simpler than the entrywise gating mechanism used in selective SSMs like Mamba (Gu & Dao, 2024). Parameter-matched baselines show substantially lower performance (LSTM: 60.8%; S4D: 81.0%; Transformer: 22.1%) (Hochreiter & Schmidhuber, 1997; Gu et al., 2022a; Vaswani et al., 2017). Among deeper two-layer variants, only S4D (96.9%) comes close to solving the task. These results suggest that explicit measure warping provides a simple and effective mechanism for selective memory when the task requires uniform allocation patterns across state dimensions.

### 3.3. Associative Memory HiPPO

Transformer architectures implement associative recall through attention mechanisms that compare queries to key–value representations of input sequences. In many modern state space models, including selective-scan architectures Mamba and Mamba-2, similar behavior emerges implicitly: learned input-dependent state updates and readouts enable content-dependent retrieval, but the intermediate key–value associations are not explicitly represented. This motivates an explicit associative memory mechanism within the HiPPO framework that achieves similar content-addressable retrieval while preserving efficiency, online updates, and direct interpretability of stored associations.

*Associative Memory HiPPO* separates temporal encoding from content-addressable storage. The HiPPO state continues to serve as a compact, structured representation of the

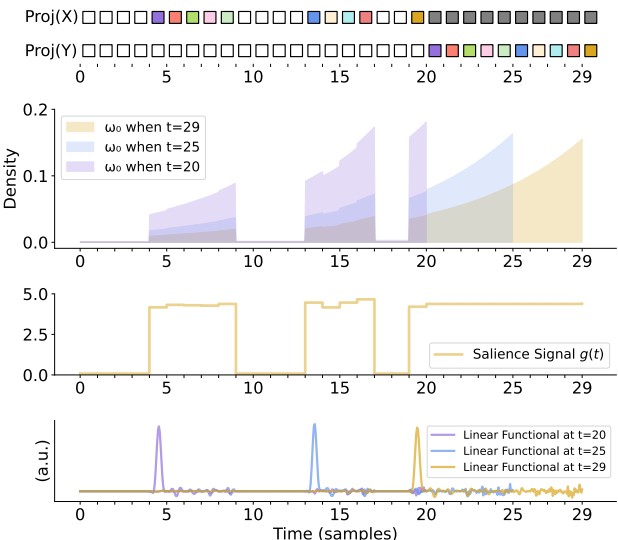

*Figure 2.* Salience HiPPO applied to a selective copying task. **Top:** Input sequence with informative tokens (colors) and uninformative tokens (white). Gray tokens indicate the recall phase, where informative tokens must be reproduced in order. **Second:** Induced history measures allocate weight to informative timepoints while downweighting uninformative timepoints. **Third:** Learned salience signal $g(t)$ decreases during uninformative inputs. **Bottom:** Output prediction functionals show peaks at corresponding informative timepoints, showing strong performance on the task.

*Table 1.* Performance on synthetic long-range memory tasks: selective copying (**SC**) and associative recall (**AR**). See Appendix C.6 for experimental details.

| Model | Params | SC Acc. | AR Acc. |
|---|---|---|---|
| *HiPPO variants (∼25k params, single-layer)* | | | |
| Vanilla | 25k | 27.6% | 22.8% |
| Vanilla + MLP | 25k | 27.7% | 20.4% |
| Salience | 25k | **100.0%** | 27.1% |
| Assoc. Memory | 25k | 65.7% | **100.0%** |
| *Baselines (single-layer, parameter-matched)* | | | |
| S4D | 26k | 81.0% | 33.2% |
| LSTM | 25k | 60.8% | 32.7% |
| Transformer | 27k | 22.1% | 33.9% |
| *Baselines (two-layer, more parameters)* | | | |
| LSTM | 48k | 78.8% | 32.0% |
| S4D | 47k | 96.9% | 33.2% |
| Transformer | 51k | 57.5% | 32.4% |

recent past, while a separate orthogonal-polynomial (OP) associative memory stores and retrieves values based on learned continuous addresses. This design allows us to expose key–value–style memory operations without entangling them with HiPPO's temporal compression.

**An Explicit OP-based Associative Memory** At each timestep, the model maintains two components. First, a HiPPO system processes the input sequence and produces a state $S_t$ that summarizes recent history. From this state, lightweight learned projections produce (i) a write address $x_{\text{key}} \in (0, 1)$ where new information should be stored, (ii) a scalar read address $x_{\text{query}} \in (0, 1)$ for reading stored values, (iii) a value vector $\mathbf{y}_t$ to be written, and (iv) scalar gates controlling when writing and reading occur. Second, the model maintains an associative memory bank consisting of $d_{\text{model}}$ independent memory channels. Each channel stores a function $m_j(x)$ on the address space $x \in [0, 1]$, represented as coefficients in an orthonormal polynomial basis. Writing updates these coefficients so that $m_j(x_{\text{key}})$ moves toward the desired value $y_t[j]$. Reading simply evaluates the memory functions at the query address: $\mathbf{r}_t[j] = m_j(x_{\text{query}})$. Because addresses are continuous and memory is represented in a polynomial basis, the OP reproducing kernel determines how values stored at nearby addresses interfere, making the similarity structure explicit and analyzable.

**Writing and Reading** Writing proceeds via a minimal-change update in coefficient space. Given a write address $x_{\text{key}}$, the current memory is evaluated to obtain $\hat{\mathbf{y}}_t[j] = m_j(x_{\text{key}})$. The coefficients are then updated with the minimum-norm change that achieves the desired value at $x_{\text{key}}$, so that $m_j(x_{\text{key}}) = \mathbf{y}_t[j]$ (see Appendix C for the closed-form expression). In function space, this corresponds to adding a localized bump proportional to the OP reproducing kernel centered at $x_{\text{key}}$. Reading evaluates the updated memory at $x_{\text{query}}$ to retrieve $\mathbf{r}_t = [m_1(x_{\text{query}}), \ldots, m_{d_{\text{model}}}(x_{\text{query}})]^\top$. This retrieved vector is passed through a lightweight output head, with a learned gate suppressing outputs outside recall phases. This mechanism mirrors key-value memory in transformers: evaluating the polynomial basis at an address produces a feature vector analogous to keys and queries, coefficient vectors store values, and the OP kernel governs similarity. Unlike attention, however, the memory is persistent, updated online, and directly interpretable through its polynomial structure.

**Associative Recall Experiment** We evaluate Associative Memory HiPPO on a synthetic associative recall task designed to isolate key-value binding and retrieval. Each episode consists of an alternating sequence of tokens drawn from two disjoint sets, followed by a single WRITE token. During the episode, associations between tokens in the first set and subsequent tokens in the second set must be stored online. At the WRITE timestep, the model is queried with a token from the first set and must retrieve the most recently associated token from the second set; the output is zero at all other timesteps. The system is trained online with truncated backpropagation through time and a mean squared loss.

Associative Memory HiPPO solves this task with a moderate number of associative coefficients and a fixed HiPPO

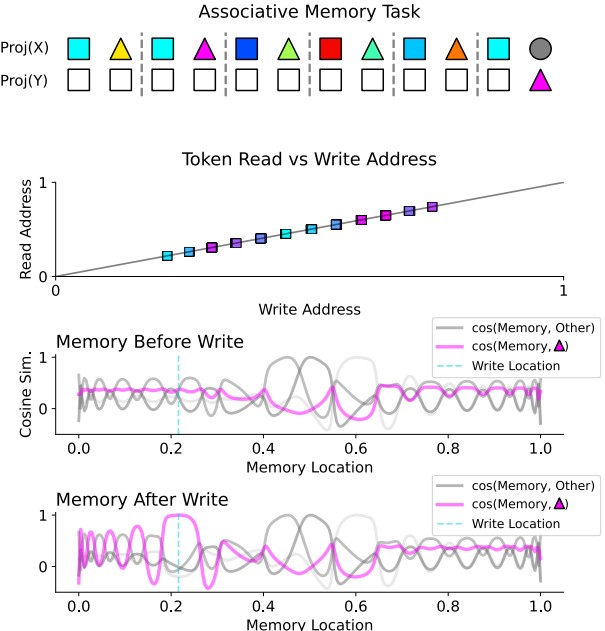

*Figure 3.* Associative Memory HiPPO uses OP associative memory to solve an associative recall task. **Top:** Task schematic. **Middle:** The system learns to read and write from consistent locations. **Bottom:** OP memory before and after a write operation.

encoder. After training, predictions at the WRITE timestep match the correct associated token, while outputs at other timesteps are suppressed by the learned gating mechanism. Crucially, the learned strategy is directly interpretable. As shown in Fig. 3, the model learns consistent write and read addresses for each association, and memory updates correspond to localized modifications of the OP memory functions. The figure visualizes the associative memory before and after a write operation, revealing how individual key–value bindings are stored and later retrieved with minimal interference. This demonstrates that content-addressable mechanisms can be realized explicitly within the HiPPO polynomial framework.

Table 1 shows that Associative Memory HiPPO achieves 100.0% test accuracy, while parameter-matched baselines fail completely: S4D, LSTM, and Transformer all remain near chance level ( 33%), even with additional depth. This suggests that linear recurrence with limited nonlinearity cannot bind arbitrary key-value pairs. Scaling investigations in Appendix C.6 show transformers require specific architectural choices (positional embeddings, depth) while LSTMs need 10-fold more parameters to reach only partial performance (68.2%), demonstrating the efficiency of task-aligned explicit mechanisms.

### 3.4. Multiscale HiPPO

The basic HiPPO system represents the recent past with a fidelity determined by a weighting function $\omega(\tau)$, produc-

ing system parameters $(A, \mathbf{b})$ and an associated orthogonal polynomial basis $\{P_n(\tau)\}_n$. A key property of HiPPO is its *timescale equivariance*: for any $g > 0$, rescaling time by $g$ corresponds to replacing $(A, \mathbf{b})$ with $(gA, g\mathbf{b})$ and evaluating the same basis at $g\tau$ (Gu et al., 2023). This property underlies the robustness of HiPPO to changes in sampling rate (Gu et al., 2020).

In many online or long-context settings, however, the relevant temporal horizon is not known in advance and may span several orders of magnitude. Rather than committing to a single timescale or a bank of independent HiPPO systems at fixed scales, we seek a single, explicit representation that can be queried across a continuum of timescales.

**Multiscale State Representation** Multiscale HiPPO achieves this by representing the HiPPO state itself as a function of an inverse timescale parameter. Concretely, we parameterize the inverse timescale as $u \triangleq \log g \in [\log \epsilon, 0]$, so that uniform resolution in $u$ corresponds to log-uniform resolution over timescales $g^{-1}$. We then represent the scale-dependent HiPPO state $\mathbf{s}(u)$ using an OP expansion,

$$\mathbf{s}(u) \approx S\,\boldsymbol{\psi}(u)\,, \quad S \in \mathbb{R}^{N \times M}\,, \tag{6}$$

where the entries of $\boldsymbol{\psi}(u)$, $\psi_0(u), \dots, \psi_{M-1}(u)$, are orthonormal polynomials on $[\log \epsilon, 0]$ with respect to the uniform weighting. Each column of $S$ corresponds to a coefficient in this scale basis, while each row corresponds to a coefficient in the HiPPO basis.

For each fixed $u$, the underlying dynamics are unchanged:

$$\dot{\mathbf{s}}(t, u) = g(u)[A\,\mathbf{s}(t, u) + \mathbf{b}f(t)], \quad g(u) = e^u. \tag{7}$$

Substituting the expansion $\mathbf{s}(u) = S\boldsymbol{\psi}(u)$ yields coupled dynamics for the coefficient matrix $S(t)$ of the form

$$\dot{S} = ASG + Bf(t)\,, \tag{8}$$

where $G \in \mathbb{R}^{M \times M}$ represents multiplication by $g(u) = e^u$ in the polynomial basis $\{\psi_m\}$, and $B$ injects the input across scales. Details of this construction, including the relationship between $G$ and the Jacobi matrix for multiplication by $u$, are given in Appendix D.

This formulation yields a *single continuous-time linear system* whose state compactly represents a continuum of effective timescales. Given a desired horizon $H$, the corresponding inverse scale $u = -\log H$ can be queried directly via $\mathbf{s}(u) = S\boldsymbol{\psi}(u)$, and decoded using the standard HiPPO orthogonal polynomial evaluations.

**Variable-Horizon Reconstruction Experiment** We evaluate Multiscale HiPPO as an explicit representation of recent history that can be queried across a wide range of horizons. We sample a one-dimensional signal as a mixture of independent Ornstein–Uhlenbeck processes with log-uniform timescales, and roll out a single Multiscale HiPPO system alongside several single-timescale HiPPO systems. At a col-

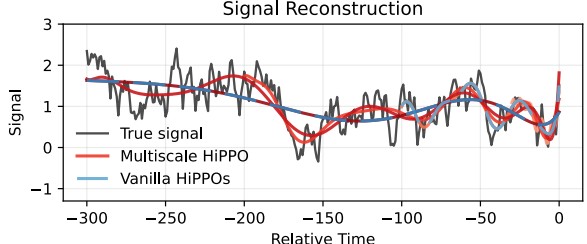

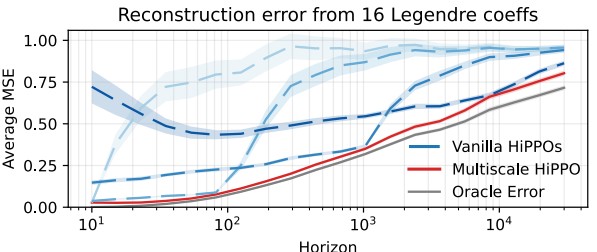

*Figure 4.* Multiscale HiPPO provides stable and parsimonious explicit representations over a continuum of timescales. **Top:** a single Multiscale HiPPO system (red) remembers the past at many timescales, while vanilla HiPPO systems (blue) remember single timescales. **Bottom:** When tasked with reproducing 16 Legendre polynomial coefficients to summarize the past at a range of timescales, Multiscale HiPPO consistently outputs coefficients of equal or better quality than single-timescale HiPPO systems (with timescales $10^1, \ldots, 10^4$, light to dark blue).

lection of horizons spanning more than three orders of magnitude, each system is tasked with producing 16 Legendre coefficients that best summarize the past over the specified interval. Single-timescale HiPPO systems are only accurate near their intrinsic horizon, while Multiscale HiPPO produces accurate reconstructions across all queried scales. As shown in Fig. 4, a single Multiscale HiPPO system matches or improves upon the reconstruction error of all single-scale baselines across horizons. This demonstrates that Multiscale HiPPO learns a parsimonious, explicit representation over a continuum of timescales, rather than committing fidelity to any single horizon.

### 3.5. Forecasting HiPPO

Modern sequence models, including SSMs used as transformer alternatives, are typically trained with one-step-ahead prediction losses. This objective implicitly determines which aspects of the past are preserved in the system state, yet the effect of this choice on memory structure is rarely made explicit. Forecasting over a finite horizon provides a natural alternative that selects different state representations. For continuous-time systems like HiPPO, this distinction is one of degree rather than kind: *Forecasting HiPPO* makes the effect of the forecasting objective explicit by constructing and visualizing horizon-dependent "predictive memories."

**A Forecasting HiPPO Mechanism** Forecasting HiPPO maintains three HiPPO systems, as illustrated in Figure 5A. System 1 represents the recent past on $[t-H, t]$ using a Leg-T system. Evaluating this representation at the left endpoint yields an estimate of the lagged signal, $u(t) \approx f(t-H)$. System 2 takes $u(t)$ as input, producing a representation of the earlier past (preceding $t-H$). During training, we learn a linear map $T$ that predicts the System 1 state from the System 2 state. At test time, System 3 is a copy of System 2 driven directly by the current input $f(t)$ instead of lagged input. Applying $T$ yields a forecast state,

$$\hat{\mathbf{s}}_{\text{future}}(t) = T\mathbf{s}_3(t),$$

which lies in the same polynomial basis as System 1 and can therefore be decoded as a prediction of the signal on the future interval $[t, t+H]$.

**Horizon-dependent Objectives Induce Geometry on Histories** A forecasting objective induces a notion of similarity on past histories. Let $h$ denote a past trajectory (the restriction of $f$ to $(-\infty, t]$), and let $\hat{f}_h(\tau)$ denote the model's prediction of the future signal at relative time $\tau \geq 0$ given history $h$. Consider a family of weighted squared-error forecasting losses

$$\mathbb{E}\left[\int_0^H w(\tau)\big(\hat{f}_h(\tau) - f(t+\tau)\big)^2 \, d\tau\right],$$

where $w(\tau)$ specifies the importance of different future times. This objective induces a positive semidefinite bilinear form on histories at time $t$,

$$\langle h, h' \rangle_Q \triangleq \mathbb{E}\left[\int_0^H w(\tau)\, \hat{f}_h(\tau)\, \hat{f}_{h'}(\tau)\, d\tau\right],$$

which defines a predictive similarity measure capturing how similarly two histories support forecasts over the horizon. In the linear Forecasting HiPPO setting, the history $h$ is encoded as a finite-dimensional coefficient vector $\mathbf{x}(h) \in \mathbb{R}^N$, and the predicted future is obtained via a linear map $\hat{\mathbf{y}}(h) = T\mathbf{x}(h)$, where $\hat{\mathbf{y}}(h)$ contains coefficients that represent $\hat{f}_h(\tau)$ in the orthogonal polynomial basis. This yields an induced quadratic form $Q = T^\top W T$ in coefficient space (see Appendix E). In our experiments we forecast in an orthonormal Legendre coefficient space with a squared Euclidean loss, so $W = I$ and hence $Q = T^\top T$. Mapping $Q$ back to the time domain by evaluating the OP basis over lags yields a history-space kernel, making explicit which portions of the past are emphasized by the objective.

**Objective-induced Predictive Memory Experiment** We apply Forecasting HiPPO to forecasting a stationary signal with multiple intrinsic timescales. We compare two models that share the same System 2/3 dynamics but differ in the System 1 horizon: a short-horizon forecaster (analogous to one-step-ahead training) and a long-horizon forecaster. Using streaming statistics, we fit the optimal linear predictor from System 2 to System 1 via reduced-rank regression, yielding an low-rank map $T$. This map identifies the sub-

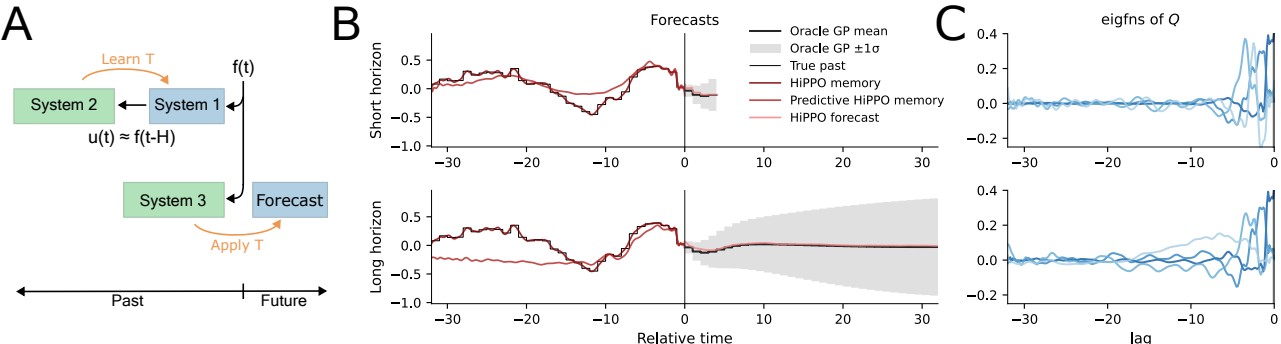

*Figure 5.* **A**: Forecasting HiPPO schematic. **B**: Forecasting HiPPO with a reduced-rank linear forecasting map reveals different "predictive memories" under different forecasting horizons. **C:** Objective-dependent history geometries in the top eigenfunctions of $Q$, the predictive history metric, show broader features for the long-horizon objective.

space of history representations that is maximally predictive of the future under the chosen objective. Projecting a history onto this subspace produces its *predictive memory*: the minimum-norm, whitened representation that preserves all information used by the rank-$d$ forecasting map. Decoding this representation back into function space allows direct visualization of which components of the past are retained or discarded. As in Figure 5B, both short- and long-horizon forecasters can accurately reconstruct the recent history, but their predictive memories differ markedly: the short-horizon forecaster emphasizes fine detail at small lags, while the long-horizon forecaster retains smoother structure extending further into the past. These differences are also reflected in the objective-induced history geometry: the leading eigenfunctions of $Q$, which identify the directions of maximal predictive similarity between histories, vary sharply near the present for short horizons and more gradually for long horizons (Figure 5C). Together, these visualizations make explicit how forecasting objectives shape memory in ways that are usually implicit in trained SSM states.

### 3.6. Evaluation on Language Modeling

The preceding sections demonstrate that HiPPO Zoo variants excel on synthetic tasks when task requirements align with explicit mechanisms. To assess performance on real-world data, we evaluate several variants on character-level language modeling using the WikiText-2 dataset.

Table 4 (Appendix F) shows that HiPPO variants underperform modern SSMs, as expected given their focus on interpretability. Vanilla HiPPO with MLP readout achieves 6.80 test perplexity versus 5.54 for Mamba (23% gap), while Salience HiPPO achieves 7.52 (36% gap). Notably, Associative Memory HiPPO performs very poorly (20.41 PPL), suggesting its continuous addressing strategy is poorly suited to autoregressive language prediction.

## 4. Discussion

The HiPPO Zoo demonstrates that capabilities characteristic of modern SSMs can be realized explicitly within an interpretable polynomial framework through principled modifications of the history measure, readout geometry, or the objective-induced metric. This enables direct visualization and analysis of memory mechanisms, but involves trade-offs in performance and computational cost.

**Interpretability vs. Performance** The Zoo prioritizes transparency over raw predictive performance. The WikiText-2 results in Section 3.6 quantify this trade-off. However, when task requirements align with explicit mechanisms, specialized variants excel (Table 1). This demonstrates the value of task-aligned explicit structure when interpretability matters as much as performance.

**Computational Trade-offs and Scalability** Each variant trades interpretability for computational resources differently. Volterra HiPPO exposes nonlinear kernels but grows as $\mathcal{O}(N^k)$ for order-$k$ systems; low-rank tensor approximations could reduce this to $\mathcal{O}(kNr)$ while preserving interpretability. Multiscale HiPPO's log-timescale formulation provides explicit coverage across orders of magnitude but costs $\mathcal{O}(NM^2)$ per update versus $\mathcal{O}(NM)$ for tridiagonal coupling. This is appropriate when unknown or variable timescales justify the overhead.

**Guidance for Practitioners** We view the HiPPO Zoo as a toolkit for choosing interpretability-performance trade-offs. Use modern SSMs (Mamba, S4D) to maximize predictive performance. Use HiPPO variants when transparency is essential: scientific or streaming applications requiring transparent memory mechanisms, or mechanistic studies of learned representations. Hybrid architectures embedding interpretable components within expressive SSMs offer a middle ground. Future work may pursue such integration or expand the zoo in orthogonal directions.

## Acknowledgments

We thank four anonymous reviewers for their helpful feedback. Research reported in this publication was supported by the National Institute of Mental Health of the National Institutes of Health under Award Number R01MH125430 and by the National Science Foundation through the Traineeship in the Advancement of Surgical Technologies, Award Number 2125528. The content is solely the responsibility of the authors and does not necessarily represent the official views of the National Institutes of Health or the National Science Foundation.

## Impact Statement

This paper presents work whose goal is to advance the field of Machine Learning. There are many potential societal consequences of our work, none of which we feel must be specifically highlighted here.

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

## A. Orthogonal Polynomial Background

Orthogonal polynomial (OP) sequences provide a classical and well-studied family of bases for representing functions with respect to a weighted inner product. Given a probability density (or more generally, a positive measure) $\omega(x)$ on a domain $\mathcal{X} \subseteq \mathbb{R}$, an orthonormal polynomial sequence $\{P_n(x)\}_{n=0}^{\infty}$ is a sequence of degree-$n$ polynomials satisfying

$$\langle P_m, P_n \rangle_\omega \triangleq \int_\mathcal{X} P_m(x) P_n(x) \omega(x) \mathrm{d}x = \delta_{mn} \ .$$

Any square-integrable function with respect to $\omega$ can be expanded in this basis, with coefficients given by inner products against the corresponding polynomials.

A fundamental structural property of OP sequences is that they satisfy a three-term recurrence relation. Favard's theorem states that a sequence of polynomials $\{P_n(x)\}_{n=0}^{\infty}$ is an orthogonal polynomial sequence with respect to some positive measure on the real line if and only if it satisfies a recurrence of the form

$$x P_n(x) = a_{n+1} P_{n+1}(x) + b_n P_n(x) + a_n P_{n-1}(x) \ , \quad a_n > 0$$

for all $n \geq 0$, with appropriate initialization. This recurrence relation underlies the structured linear operators that appear in HiPPO dynamics and enable their computational efficiency.

Different choices of the weighting function $\omega$ induce different orthogonal polynomial families and corresponding notions of approximation error. Classical families such as Legendre, Laguerre, and Chebyshev polynomials correspond to different weighting schemes and lead to distinct memory behaviors when used within HiPPO-style systems. In this work, we focus on a specific orthogonal polynomial family that admits efficient structured dynamics and well-conditioned numerical behavior: the Legendre polynomials. Below, we define the Legendre polynomials, while explicit forms of the HiPPO systems used in this paper are listed in Table 2. We refer the reader to the survey of Totik and the book of Chihara for more information on orthogonal polynomials (Totik, 2005; Chihara, 1978).

**Orthonormal Legendre polynomials on** $[0, 1]$**.** Let $P_n(x)$ denote the standard Legendre polynomials on $[-1, 1]$, defined by the recurrence

$$(n + 1) P_{n+1}(x) = (2n + 1) x P_n(x) - n P_{n-1}(x), \qquad P_0(x) = 1, \ P_1(x) = x,$$

and orthogonal with respect to the uniform weighting on $[-1, 1]$. Throughout this paper we use a shifted and rescaled version that is orthonormal on $[0, 1]$,

$$L_n(s) \ \triangleq \ \sqrt{2n + 1} \, (-1)^n \, P_n(2s - 1), \qquad s \in [0, 1].$$

These polynomials satisfy

$$\int_0^1 L_n(s) \, L_m(s) \, \mathrm{d}s = \delta_{nm},$$

and are oriented so that $s = 0$ corresponds to the present and $s = 1$ corresponds to the furthest point in the represented past.

## B. HiPPO Systems Table

For completeness, Table 2 summarizes the continuous-time HiPPO systems used throughout the paper. We focus on two Legendre-based constructions that differ only in their choice of weighting function over the past. Both systems admit closed-form expressions for the HiPPO dynamics matrices $(A, \mathbf{b})$ and an orthonormal polynomial basis, enabling stable and efficient continuous-time implementations.

## C. Experimental Details

The following sections provide experimental details necessary to reproduce the results in this paper. Code capable of reproducing these results is provided at: https://github.com/jackgoffinet/hippo-zoo

### C.1. Quadratic Volterra Example

**Task and Data** We evaluate Volterra HiPPO on the nonlinear benchmark system of Wray & Green (1994), described below, which admits a purely second-order Volterra representation. The input $f(t)$ is a band-limited white noise signal generated with sampling step $\Delta t = 10^{-2}$ and cutoff frequency 10. The target output $y(t)$ is produced by applying the Wray-Green

|        | $\omega(\tau)$ | $P_n(\tau)$ | $b_n$ | $A_{nk}$ |
|--------|----------------|-------------|-------|----------|
| Leg-T  | $\mathcal{U}[0,1]$ | $L_n(\tau)$ | $\sqrt{2n+1}$ | $\begin{cases} -b_n b_k (-1)^{n-k}, & n > k \\ -b_n b_k, & n = k \\ 0, & n < k \end{cases}$ |
| Leg-S  | $\exp(-\tau)$ | $L_n(1 - e^{-\tau})$ | $\sqrt{2n+1}$ | $\begin{cases} 0, & n < k \\ -b_n b_k \dfrac{n+1}{2n+1}, & n = k \\ -b_n b_k, & n > k \end{cases}$ |

*Table 2.* Orthogonal Polynomial HiPPO systems used in this paper. Leg-T corresponds to the truncated Legendre HiPPO system with uniform weighting on a finite window, while Leg-S corresponds to the exponentially weighted (sliding) Legendre HiPPO system.

system to the input stream, yielding a sequence $\{(f_t, y_t)\}_{t=1}^T$ with $T = 10^7$.

The nonlinear benchmark introduced by Wray & Green (1994) is a purely second-order Volterra system with a separable quadratic kernel. Let $f(t)$ denote the input and define the impulse response

$$\mu(\tau) = \frac{a}{m} e^{-k\tau} \sin(m\tau),$$

where $a, m, k > 0$ are fixed constants. The output is given by a quadratic convolution

$$y(t) = \int_0^{\tau_{\max}} \int_0^{\tau_{\max}} h_2(\tau_1, \tau_2) \, f(t - \tau_1) \, f(t - \tau_2) \, d\tau_1 d\tau_2,$$

with separable kernel

$$h_2(\tau_1, \tau_2) = \alpha \, \mu(\tau_1) \, \mu(\tau_2),$$

where $\alpha$ is a scaling constant. Equivalently, letting

$$z(t) = \int_0^{\tau_{\max}} \mu(\tau) \, f(t - \tau) \, d\tau,$$

the system reduces to $y(t) = \alpha \, z(t)^2$. Thus, the Wray-Green system has no linear term and admits an exact second-order Volterra representation. In our implementation, $a = 2$, $m = 0.3$, $k = 0.08$, $\tau_{\max} = 50$, and $\alpha = 0.004$.

**HiPPO Memory** We maintain a HiPPO memory state $\mathbf{s}_t \in \mathbb{R}^N$ with $N = 64$ using a scaled Legendre HiPPO system (Leg-S). Let $(A, \mathbf{b})$ denote the continuous-time HiPPO parameters for Leg-S. We rescale the system by a constant timescale factor $\alpha = 0.375$ using $A \leftarrow A/\alpha$ and $\mathbf{b} \leftarrow \mathbf{b}/\alpha$. We then discretize the dynamics with zero-order hold:

$$A_d = \exp(\Delta t A) \,, \quad \mathbf{b}_d = A^{-1}(A_d - I)\mathbf{b} \,,$$

and evolve the state online as $\mathbf{s}_{t+1} = A_d \mathbf{s}_t + \mathbf{b}_d f_t$. During training, the HiPPO dynamics are fixed. Only the readout parameters are learned.

**Readouts (linear, quadratic, and MLP)** We compare three predictors trained online from the shared state $\mathbf{s}_t$:

- **Linear Volterra readout:** $\hat{y}_t = b + \mathbf{w}_1^\top \mathbf{s}_t$

- **Quadratic Volterra readout:** $\hat{y}_t = b + \mathbf{w}_1^\top \mathbf{s}_t + \mathbf{s}_t^\top W_2 \mathbf{s}_t$, where $W_2 \in \mathbb{R}^{N \times N}$ is an unconstrained quadratic coefficient matrix.

- **MLP readout:** a single-hidden-layer network with $\hat{y}_t = b_0 + W_o \tanh(W_h \mathbf{s}_t + b_h)$ with hidden width 128.

**Training Protocol** All three predictors are trained by online stochastic gradient descent on the squared error $(\hat{y}_t - y_t)^2$. For the linear and quadratic readouts, we use SGD with a learning rate $3 \times 10^{-2}$, and for the MLP, we use SGD with a learning rate $10^{-2}$. We report cumulative error $\sum_{s=1}^t (\hat{y}_s - y_s)^2$ as a function of time $t$ on log-log axes.

**True Kernel Visualization** To visualize the ground-truth second-order Volterra kernel used in the experiment, we plot a discretized kernel $h_2(\tau_1, \tau_2)$ over $\tau_1, \tau_2 \in \{0, \ldots, 49\}$ using $h_2(\tau_1, \tau_2) \propto \mu(\tau_1)\mu(\tau_2)$ with $\mu(\tau) = \frac{a}{m} e^{-k\tau} \sin(m\tau)$ and constants $a = 2.0$, $m = 0.3$, and $k = 0.08$. The result is scaled by $4 \times 10^{-3}$ to match the Wray-Green system implementation.

**Inferred Kernel from the Quadratic Readout** To interpret the learned quadratic coefficients $W_2$ in the time-lag domain, we reconstruct the implied second-order kernel on a grid $\tau \in \{0, \ldots, 49\}\Delta t$. Let $\{P_n\}_{n=0}^{N-1}$ denote the Legendre HiPPO system. We form basis evaluations $P_n(\tau/\alpha)$ and combine them with the learned coefficients and the induced lag-density $p(\tau)$ associated with the scaled timescale, $p(\tau) \propto \frac{\Delta t}{\alpha} \exp(-\tau/\alpha)$, to obtain an inferred kernel of the form

$$\hat{h}_2(\tau_1, \tau_2) \propto p(\tau_1)p(\tau_2) \sum_{i,j=0}^{N-1} (W_2)_{ij} P_i(\tau_1/\alpha) P_j(\tau_2/\alpha) \,,$$

which is then visualized with a symmetric diverging colormap.

## C.2. Selective Copying Experiment

**Task and Data** We evaluate Salience HiPPO on an iterative selective copying task with vector-valued tokens (18 unique tokens: 1 WRITE token, 1 uninformative token, and 16 informative tokens, all uniformly randomly drawn unit norm tokens in $\mathbb{R}^{32}$). Each episode has length $T = 30$ and consists of two phases. In the first phase, 10 informative tokens are randomly interleaved with 10 uninformative tokens. In the final 10 timesteps (the write phase), the special write token is presented at every timestep and the model is required to output the informative tokens in the order in which they were presented.

**HiPPO Memory** The model maintains a multi-channel HiPPO memory $S_t \in \mathbb{R}^{d_{\text{model}} \times N}$, $d_{\text{model}} = 64$ channels and $N = 256$ coefficients per channel, using the Leg-S HiPPO system. Concretely, this consists of $d_{\text{model}}$ independent and identical HiPPO systems (one for each channel) with no cross-channel interactions. The continuous-time dynamics are discretized with a zero-order hold scheme. The HiPPO parameters $(A, \mathbf{b})$ are fixed throughout training, scaled by the episode length to match the task timescale.

**Salience Parameterization** The salience signal $g(t) \in (0, g_{\text{max}})$ is a learned scalar produced at each timestep. The current input token is first projected into the model dimension, yielding $x_t^{\text{proj}} \in \mathbb{R}^{d_{\text{model}}}$. In parallel, a low-dimensional summary of the current memory state is computed by applying a learned linear map to each channel of $S_t$, passing through a $\tanh$ nonlinearity, and averaging across channels. The concatenation of $x_t^{\text{proj}}$ and this pooled memory summary is passed through a small MLP with a single hidden layer of width 128, softplus hidden activation, and a scaled sigmoid output activation to produce $g(t)$. This scalar modulates the effective integration rate of the HiPPO dynamics, yielding an input- and state-dependent deformation of the history measure.

**Training Protocol** Models are trained online using truncated backpropagation through time (TBPTT). The concatenated episodes are processed sequentially and divided into chunks of length $T$, the episode length. Gradients are computed within each chunk, and the HiPPO state is detached between chunks. All neural parameters (input projection, salience network, and readout) are optimized with AdamW using a learning rate of $10^{-3}$.

**Interpretability Diagnostics** For evaluation, we record the learned salience trace $g(t)$ over an episode and compute the induced warped-time map $\varphi(t) = \int_0^t g(s)\mathrm{d}s$ under the assumption that $g$ is piecewise constant between discrete timesteps. Using this map, we visualize the induced history measures in real time and the effective linear functionals applied to history during the write phase. These diagnostics are used to generate the plots in Fig. 2, illustrating how selective copying is implemented through explicit reshaping of the history measure.

## C.3. Associative Recall Experiment

**Task and Data** We evaluate Associative Memory HiPPO on a synthetic associative recall task designed to isolate online key–value binding and retrieval. Each episode has even length $T = 12$ and consists of alternating tokens from two disjoint sets $A = \{a_1, \ldots, a_{12}\}$ and $B = \{b_1, \ldots, b_{12}\}$ followed by a final WRITE token. Concretely, timesteps $t = 0, 2, \ldots, T-2$ present $A$ tokens, timesteps $t = 1, 3, \ldots, T-3$ present $B$ tokens, and $t = T-1$ presents WRITE. The query token at time $T-2$ is an $A$ token that is guaranteed to have appeared previously among the $A$ positions. The target output at time $T-1$ is the $B$ token that followed the most recent prior occurrence of this queried $A$ token; outputs are zero at all other timesteps. Inputs and outputs are represented as normalized vectors in $\mathbb{R}^{24}$ using a fixed random token table.

**Fixed-ZOH HiPPO Encoder** We maintain a HiPPO state $S_t \in \mathbb{R}^{d_{\text{model}} \times n_{\text{hippo}}}$ updated independently per channel, where $d_{\text{model}} = 32$ and $n_{\text{hippo}} = 32$. The input $x_t \in \mathbb{R}^{24}$ is mapped to channels via a learned linear projection $u_t = W_{\text{in}}x_t \in \mathbb{R}^{d_{\text{model}}}$. Using a single precomputed zero-order-hold (ZOH) discretization of a Leg-T HiPPO scheme and a fixed timescale

2.0, the recurrence is

$$S_{t+1}[j] = A_d S_t[j] + b_d u_t[j], \qquad j = 1, \ldots, d_{\text{model}},$$

where $S_t[j] \in \mathbb{R}^{n_{\text{hippo}}}$ denotes the $j$-th channel state.

**OP Associative Memory State** In addition to $S_t$, we maintain an associative memory bank

$$C_t \in \mathbb{R}^{d_{\text{model}} \times n_{\text{assoc}}},$$

where $n_{\text{assoc}} = 32$. Each row $C_t[j]$ represents the coefficients of a function $m_j(\cdot; t)$ on a continuous address space $x \in [0, 1]$ using an orthonormal Legendre basis $\phi(x) \in \mathbb{R}^{n_{\text{assoc}}}$ on $[0, 1]$:

$$m_j(x; t) = \langle C_t[j], \phi(x) \rangle.$$

**Addressing, Gates, and Value Signal** From the updated HiPPO state $S_{t+1}$, lightweight networks produce: (i) a write address $x_{\text{key}} \in (0, 1)$ and (ii) a read address $x_{\text{query}} \in (0, 1)$, both via learned linear projections with a sigmoid activation, and (iii) a write gate $g_{\text{write}} \in (0, 1)$ and (iv) an output gate $g_{\text{out}} \in (0, 1)$, both via two-layer MLPs with hidden width 256, tanh hidden activations, a residual connection, and sigmoid output activations. The value signal written to memory is a learned linear map of the channel input, $y_t \in \mathbb{R}^{d_{\text{model}}}$.

**Minimum-norm Write Update (Orthogonal Projection)** Let $k_t = \phi(x_{\text{key}}) \in \mathbb{R}^{n_{\text{assoc}}}$ and define the current memory evaluation (per channel)

$$\hat{y}_t[j] = \langle C_t[j], k_t \rangle, \qquad j = 1, \ldots, d_{\text{model}}.$$

We update $C_t$ to move $\hat{y}_t$ toward the desired value $y_t$ while changing coefficients as little as possible in the $\ell_2$ norm. Specifically, for each channel $j$ we solve

$$\Delta C_t[j] = \arg\min_{\Delta \in \mathbb{R}^{n_{\text{assoc}}}} \|\Delta\|_2^2 \quad \text{s.t.} \quad \langle C_t[j] + \Delta, k_t \rangle = (1 - g_{\text{write}}) \hat{y}_t[j] + g_{\text{write}} y_t[j].$$

This constraint interpolates between no write ($g_{\text{write}} = 0$) and a full write ($g_{\text{write}} = 1$). The solution follows from a Lagrange multiplier and yields the closed-form update

$$C_{t+1}[j] = C_t[j] + \alpha_t (y_t[j] - \hat{y}_t[j]) k_t, \qquad \alpha_t = \frac{g_{\text{write}}}{\|k_t\|_2^2 + \varepsilon}.$$

Because the basis is orthonormal, $\|k_t\|_2^2 = \langle k_t, k_t \rangle$ is well-conditioned. Interpreted in function space, the update above adds a localized bump proportional to the truncated reproducing kernel $K(x_{\text{key}}, x) = \phi(x_{\text{key}})^\top \phi(x)$, making interference between different addresses explicit. Figure 6 illustrates the kernel structure and learned address spacing for $n_{assoc} = 16$ coefficients.

**Reading and Output** Reading evaluates the updated memory at the query address:

$$r_t[j] = \langle C_{t+1}[j], q_t \rangle, \qquad q_t = \phi(x_{\text{query}}),$$

forming a retrieved vector $r_t \in \mathbb{R}^{d_{\text{model}}}$. A linear output projection maps to token space and is gated:

$$\hat{x}_t = g_{\text{out}} W_{\text{out}} r_t \in \mathbb{R}^{d_{\text{in}}}.$$

**Training Protocol** Training proceeds online: each iteration samples one episode, initializes $(S_0, C_0) = (0, 0)$, and rolls out the dynamics sequentially. We use truncated backpropagation through time (TBPTT) with chunk length equal to the episode length in our experiments, applying a stop-gradient operation to $(S, C)$ at chunk boundaries. Parameters are optimized with AdamW with a learning rate $10^{-3}$ to minimize mean squared error between $\hat{x}_t$ and the target output at the WRITE timestep and zero elsewhere.

### C.4. Variable-Horizon Reconstruction Experiment

**Data** We generate a one-dimensional stationary signal $x_{0:T-1}$ as an equal-weight mixture of $K$ independent Ornstein–Uhlenbeck (OU) components with log-uniform time constants. Concretely, for $\Delta t = 1$ and component timescales $\tau_k \sim \text{LogUniform}(2, 2000)$, each component evolves as an AR(1)

$$z_{t+1}^{(k)} = a_k z_t^{(k)} + b_k \varepsilon_t^{(k)}, \qquad a_k = \exp(-\Delta t / \tau_k), \qquad b_k = \sqrt{1 - a_k^2},$$

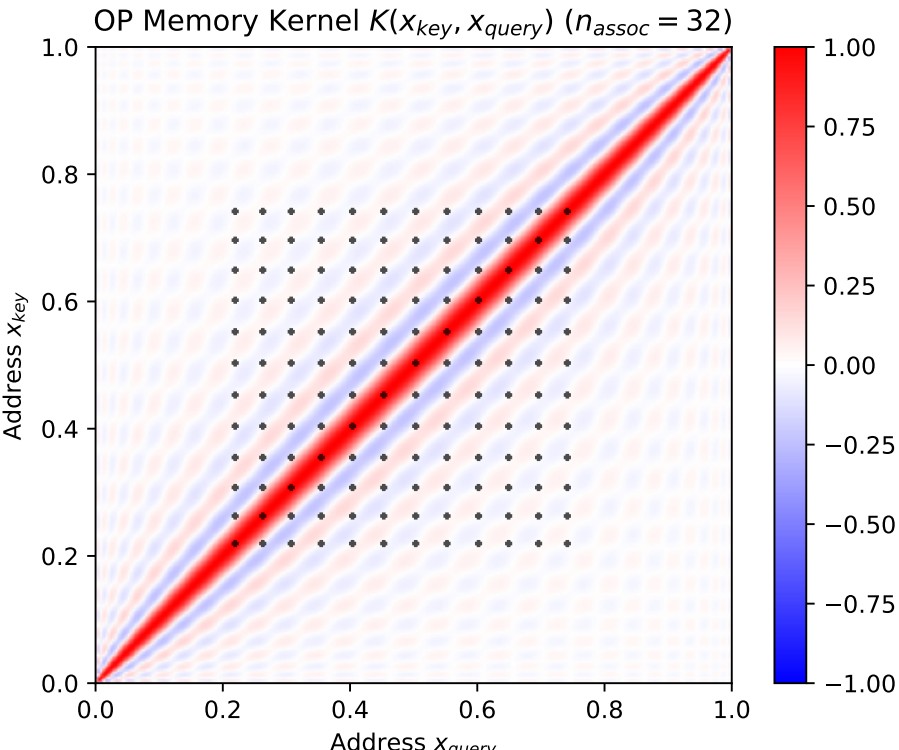

*Figure 6.* Reproducing kernel $K(x_{key}, x_{query})$ of the Legendre OP basis with $n_{assoc} = 32$ memory coefficients, evaluated over the address space $x \in [0, 1]$. Diagonal entries equal 1 by definition ($\kappa(x, x) = 1$). Learned write addresses (markers) are spaced to minimize off-diagonal kernel values, resulting in near-zero interference between stored associations.

with $\varepsilon_t^{(k)} \sim \mathcal{N}(0, 1)$ and $z_0^{(k)} \sim \mathcal{N}(0, 1)$. The observed signal is

$$x_t = \frac{1}{\sqrt{K}} \sum_{k=1}^{K} z_t^{(k)},$$

which has unit stationary variance. We report results averaged over 64 independently sampled trajectories of length $T = 3 \times 10^4$.

**Multiscale and Single-Scale Systems** All systems are run online on the full sequence to produce a final state at time $T - 1$. We compare: (i) three Multiscale HiPPO variants (basic $g$-polynomial, Jeffreys-weighted $g$-polynomial, and log-timescale $u = \log g$; see Appendix D for complete descriptions), each with HiPPO order $n = 16$ and scale dimension $m = 128$, so the total state has dimension $nm$; and (ii) a bank of vanilla HiPPO systems at fixed base timescales $\tau \in \{10, 100, 1000, 10000\}$, each storing $n = 16$ coefficients. All systems use the same underlying continuous-time HiPPO Leg-T HiPPO system and are discretized with a zero-order hold (ZOH) step of $\Delta t = 1$.

For the multiscale systems, we implement the coupled linear dynamics

$$\dot{S}(t) = A\, S(t)\, M + B\, x(t),$$

where $S(t) \in \mathbb{R}^{n \times m}$ is the multiscale state, $A \in \mathbb{R}^{n \times n}$ is the HiPPO generator (scaled by a base timescale parameter), and $M \in \mathbb{R}^{m \times m}$ is the scale-coupling matrix (tridiagonal for polynomial-in-$g$ variants; dense for the $u = \log g$ variant). In code we vectorize $S$ and discretize the resulting linear system exactly using ZOH to obtain a discrete-time update

$$\mathbf{s}_{t+1} = A_d \mathbf{s}_t + B_d x_t,$$

where $\mathbf{s}_t = \text{vec}(S(t)) \in \mathbb{R}^{nm}$.

**Querying a Target Horizon**  To evaluate reconstruction at horizon $L \in \{L_1, \ldots, L_R\}$, we interpret each system's output as $n = 16$ orthonormal shifted Legendre coefficients on a normalized coordinate $s \in [0, 1]$ spanning the requested window. We fix the convention

$$s = 0 \leftrightarrow \text{present time } (T - 1), \qquad s = 1 \leftrightarrow \text{past endpoint } (T - 1 - L), \qquad t(s) = (T - 1) - sL.$$

Given coefficients $c(L) \in \mathbb{R}^n$ in the orthonormal shifted Legendre basis $\{\tilde{P}_k(s)\}_{k=0}^{n-1}$ on $[0, 1]$, the reconstructed curve is

$$\hat{x}_L(s) = \sum_{k=0}^{n-1} c_k(L) \, \tilde{P}_k(s).$$

We evaluate $\hat{x}_L$ on a uniform grid of $R = 128$ points in $s \in [0, 1]$.

*Multiscale Systems* For each horizon $L$, the multiscale state is queried by evaluating the scale-basis vector at an inverse-horizon parameter and contracting along the scale dimension:

$$c(L) = S(T - 1) \, q(L),$$

where $q(L) \in \mathbb{R}^m$ is the scale-basis evaluated at the appropriate scale. In the basic $g$-polynomial construction, we use

$$g(L) = \text{clip}\Big(\frac{\tau_0}{L}, \, 0, \, 1\Big), \qquad q(L) = \boldsymbol{\ell}\big(g(L)\big),$$

with $\tau_0$ the multiscale base timescale used to scale $A$ and $\boldsymbol{\ell}$ the orthonormal shifted Legendre basis on $[0, 1]$. For the Jeffreys-weighted $g$ variant, the basis is replaced by orthonormal polynomials for $\omega(g) \propto 1/g$ on $[\epsilon, 1]$ and we clip $g(L)$ into $[\epsilon, 1]$. For the log-timescale $u = \log g$ variant, we set

$$g(L) = \text{clip}\Big(\frac{\tau_0}{L}, \, \epsilon, \, 1\Big), \qquad u(L) = \log g(L) \in [\log \epsilon, 0], \qquad q(L) = \boldsymbol{\psi}\big(u(L)\big),$$

where $\boldsymbol{\psi}$ is an orthonormal polynomial basis that is uniform in $u$.

*Vanilla HiPPO Systems* Each fixed-timescale HiPPO system with base window $\tau$ produces coefficients $c^{(\tau)} \in \mathbb{R}^n$ that summarize the past over a fixed window of length $\tau$, with its own normalized coordinate $s_\tau \in [0, 1]$ defined by

$$t(s_\tau) = (T - 1) - s_\tau \tau.$$

To compare fairly at requested horizon $L$, we evaluate the vanilla system on the *same absolute-time grid* $t(s) = (T-1) - sL$ used above, but map each query time into the system's coordinate

$$s_\tau(t) = \frac{(T - 1) - t}{\tau} = \frac{sL}{\tau},$$

and set predictions to zero when $s_\tau(t) \notin [0, 1]$ (i.e., when the requested time lies outside the system's representable window). This implements the behavior that fixed-horizon HiPPO cannot represent content earlier than its intrinsic window.

**Reconstruction error.**  For each horizon $L$, we define the mean squared error (MSE) by comparing $\hat{x}_L(s)$ to the true signal sampled at the corresponding (possibly fractional) times $t(s)$ via linear interpolation:

$$\text{MSE}(L) = \frac{1}{R} \sum_{i=1}^{R} \Big(\hat{x}_L(s_i) - x\big(t(s_i)\big)\Big)^2, \qquad s_i = \frac{i - 1}{R - 1}.$$

We evaluate $\text{MSE}(L)$ for each trajectory and report the Monte Carlo mean across trajectories; shaded bands in the plots correspond to the standard error of the mean across the 64 trials.

**Reported Curves and Example Reconstructions**  The top panel of Fig. 4 plots average $\text{MSE}(L)$ versus $L$ (logarithmic $x$-axis) for the log-timescale Multiscale HiPPO system and each fixed-timescale vanilla HiPPO baseline. The bottom panel shows representative reconstructions $\hat{x}_L$ for several horizons $L$ on a single example trajectory, illustrating that the multiscale representation yields accurate reconstructions over a wide range of horizons under a fixed HiPPO coefficient budget.

### C.5. Objective-induced Predictive Memory Experiment

We provide additional details for the Forecasting HiPPO experiment in Section 3.5, including the system construction, training procedure, and evaluation protocol.

**Signal Model** The input signal $x(t)$ is drawn from a stationary zero-mean Gaussian process constructed as a mixture of independent RBF kernels with different lengthscales,

$$\kappa(\Delta) = \sum_{m=1}^{M} w_m^2 \exp\left(-\frac{\Delta^2}{2\ell_m^2}\right),$$

where the weights $\{w_m\}$ are normalized so that $\sum_m w_m^2 = 1$, ensuring $\text{Var}[x(t)] \approx 1$. The experimental lengthscales are $(10^{-1},\ 3,\ 16,\ 32,\ 64)$ while the corresponding weights are $(\frac{1}{2},\ 2,\ 5,\ 5,\ 5)$. Samples are generated on an integer grid using a circulant embedding and FFT-based approximation. The same realization of $x(t)$ is used for both short- and long-horizon forecasters.

**HiPPO Systems and Discretization** Each forecaster maintains three continuous-time Leg-T HiPPO systems with $N = 64$ coefficients, discretized with an exact zero-order hold (ZOH).

- **System 1 (Recent Past)** encodes the signal over a horizon $H$ using a Legendre HiPPO system on the interval $[t - H, t]$.

- **System 2 (Past-before-Past)** integrates a scalar lagged signal extracted from System 1, approximating $x(t - H)$, and therefore represents the interval preceding the recent window.

- **System 3 (Aligned Past)** has identical dynamics to System 2, but is driven directly by the true input $x(t)$, allowing it to encode the recent window in the same coordinate system as System 2.

Systems 2 and 3 have timescale of 32, while System 1 has a timescale of $H = 4$ for the short horizon forecaster, and $H = 32$ for the long horizon forecaster.

**Learning the Forecast Map** Forecasting is posed as a linear prediction problem between HiPPO states. Let

$$x_t = S_2(t), \qquad y_t = S_1(t),$$

where $x_t$ represents history preceding the forecast window and $y_t$ represents the recent window state to be predicted.

During a single online pass over the training signal of length $T = 2 \times 10^4$, we accumulate second-order statistics

$$\Sigma_{xx} = \mathbb{E}[x_t x_t^\top], \quad \Sigma_{yy} = \mathbb{E}[y_t y_t^\top], \quad \Sigma_{yx} = \mathbb{E}[y_t x_t^\top],$$

using simple running averages.

We then compute a rank-$d$ reduced-rank regression (RRR) predictor

$$\hat{y}_t = T_d x_t,$$

where $T_d$ minimizes $\mathbb{E}\|y_t - T x_t\|^2$ subject to $\text{rank}(T) \leq d$. This is obtained via an SVD of the whitened cross-covariance, as detailed in Appendix E. The same procedure is applied independently for short- and long-horizon forecasters, which differ only in the System 1 horizon $H$.

**Predictive Memory and Projection** In addition to the forecast map $T_d$, the RRR solution yields a rank-$d$ projector

$$P_x : \mathbb{R}^n \to \mathbb{R}^n,$$

which projects System 3 states onto the subspace of history representations that are maximally predictive of the future over the chosen horizon. We refer to $P_x S_3(t)$ as the *predictive memory* associated with history $h$.

Both the full System 3 state and its predictive projection are decoded back into function space using the standard HiPPO evaluation map, allowing direct visualization of which portions of the past are retained or discarded by the forecasting objective.

As a reference for the HiPPO forecasts, we additionally compute the Gaussian process posterior mean and uncertainty conditioned on the same observed history. All plots use a zero-order-hold (piecewise constant) convention consistent with the discretized HiPPO dynamics.

To characterize the geometry induced by the forecasting objective, we compute the operator

$$Q = T^\top T,$$

and map it back to the time domain by evaluating the orthogonal polynomial basis on a dense lag grid, yielding a history-space kernel whose leading eigenfunctions are shown in Figure 5.

### C.6. Selective Copying and Associative Recall Baseline Experiments

We compare HiPPO Zoo variants to standard sequence models on the synthetic copying and associative recall tasks described above. All models are trained for 4,000 gradient steps using AdamW (weight decay $10^{-4}$) on 3,500 sequences (70% train, 15% validation, 15% test). Learning rates from $\{3 \times 10^{-4}, 10^{-3}, 3 \times 10^{-3}\}$ are selected via validation loss. Single-layer baselines are parameter-matched to isolate memory mechanisms. We additionally present multilayer baselines as representative "standard" configurations. Test accuracies are calculated using nearest token retrieval. Results are presented in Table 1.

Due to the poor performance of the baseline models, we performed additional scaling investigations for the associative recall (AR) task. Results are shown in Table 3. First, we note that single-layer transformers cannot solve the task regardless of capacity, data, or steps. However, we find two-layer transformers can solve the task with more steps or more capacity and positional embeddings, which is intuitive given the task structure (find the most recent occurrence of the query token, then select the next token). Second, LSTMs only achieve ¿60% accuracy with 10× capacity, 10× data, and 2× steps. Still, the accuracy is far from 100%, unlike high-performing transformer settings. Third, we find that S4D is unable to solve AR in any of the tested settings, suggesting that linear recurrence with limited nonlinearities may be unable to bind arbitrary token pairs. In contrast, Associative Memory HiPPO is able to achieve 100% test accuracy in its baseline configuration, demonstrating the advantage of task-aligned explicit structure.

*Table 3.* Follow-up associative recall (AR) experiments investigating the effects of model scale, dataset size, optimization budget, and positional embeddings (PEs). Baseline settings correspond to the original AR experiments. "Capacity 10×" increases model width to approximately 10× the original parameter count, "Data 10×" increases the number of training sequences by 10×, and "Steps 2×" doubles the number of optimization steps.

| Setting | Model | # Layers | Params | Test Acc. |
|---|---|---|---|---|
| *HiPPO Zoo baseline* | | | | |
| Baseline | Assoc. Memory HiPPO | 1 | 25k | **100.0%** |
| *Transformer variants* | | | | |
| Baseline | Transformer | 1 | 27k | 32.9% |
| Baseline + PE | Transformer | 1 | 27k | 33.2% |
| Capacity 10× | Transformer | 1 | 270k | 32.4% |
| Capacity 10× + PE | Transformer | 2 | 270k | 98.9% |
| Data 10× | Transformer | 1 | 27k | 32.9% |
| Data 10× + PE | Transformer | 2 | 27k | 32.7% |
| Steps 2× | Transformer | 1 | 27k | 34.5% |
| Steps 2× + PE | Transformer | 2 | 27k | 99.7% |
| Capacity 10× + Data 10× + Steps 2× | Transformer | 1 | 270k | 32.3% |
| Capacity 10× + Data 10× + Steps 2× | Transformer | 2 | 270k | 31.2% |
| *LSTM and S4D* | | | | |
| Baseline | LSTM | 1 | 25k | 32.7% |
| Baseline | LSTM | 2 | 48k | 32.0% |
| Baseline | S4D | 1 | 26k | 33.2% |
| Baseline | S4D | 2 | 47k | 33.2% |
| Capacity 10× + Data 10× + Steps 2× | LSTM | 1 | 250k | 68.2% |
| Capacity 10× + Data 10× + Steps 2× | LSTM | 2 | 480k | 77.8% |
| Capacity 10× + Data 10× + Steps 2× | S4D | 1 | 260k | 32.8% |
| Capacity 10× + Data 10× + Steps 2× | S4D | 2 | 470k | 32.8% |

## D. Multiscale HiPPO: Additional Derivations and Details

This appendix provides additional technical details for Multiscale HiPPO that are omitted from the main text. We first derive the basic multiscale construction using polynomial embeddings of the inverse timescale parameter $g$. We then discuss Jeffreys-style scale embeddings and explain why they do not alter the limiting spectral properties of the resulting system. Finally, we give further details for the log-timescale formulation $u = \log g$, which is the variant used in the main text

experiments. Figure 7 compares the three methods on the variable-horizon reconstruction experiment in Section 3.4. +

**D.1. Basic Multiscale HiPPO via Polynomial Embeddings in $g$**

Recall that for a fixed inverse timescale $g > 0$, the continuous-time HiPPO dynamics are

$$\dot{\mathbf{s}}(t; g) = g\big[A\,\mathbf{s}(t; g) + \mathbf{b}\,f(t)\big],$$

where $\mathbf{s}(t; g) \in \mathbb{R}^N$ is the vector of HiPPO coefficients at scale $g$. To represent a continuum of such states, we assume that $\mathbf{s}(t; g)$ admits an expansion in an orthonormal polynomial basis on $g \in [0, 1]$,

$$\mathbf{s}(t; g) \;\approx\; S(t)\,\boldsymbol{\ell}(g), \qquad \boldsymbol{\ell}(g) = \big(L_0(g), \dots, L_{M-1}(g)\big)^\top,$$

where $\{L_m\}$ are shifted Legendre polynomials orthonormal on $[0, 1]$ and $S(t) \in \mathbb{R}^{N \times M}$ collects the scale coefficients.

Substituting this ansatz into the HiPPO dynamics gives

$$\dot{S}(t)\,\boldsymbol{\ell}(g) = gAS(t)\,\boldsymbol{\ell}(g) + g\mathbf{b}\,f(t).$$

Multiplication by $g$ acts linearly on the polynomial basis, so there exists a matrix $C \in \mathbb{R}^{M \times M}$ such that

$$g\,\boldsymbol{\ell}(g) = C\,\boldsymbol{\ell}(g).$$

For orthonormal polynomials on a compact interval, Favard's theorem guarantees that $C$ is a symmetric tridiagonal matrix (see Appendix A for a statement of Favard's theorem). Choosing

$$B \in \mathbb{R}^{N \times M} \quad \text{such that} \quad B\,\boldsymbol{\ell}(g) = g\,\mathbf{b},$$

we obtain the multiscale dynamics

$$\dot{S} = ASC + Bf(t).$$

Vectorizing the state yields

$$\mathrm{vec}(\dot{S}) = (C^\top \otimes A)\,\mathrm{vec}(S) + \mathrm{vec}(B)\,f(t),$$

and since $C$ is symmetric, its eigenvalues $\{\lambda_m\}$ lie in $[0, 1]$. The spectrum of the full system consists of products $\lambda_m \mu_n$ with $\mu_n \in \mathrm{eig}(A)$, corresponding to a bank of HiPPO systems at effective timescales indexed by $\lambda_m$.

**D.2. Jeffreys-Style Scale Embeddings**

A natural idea is to bias the multiscale representation toward long horizons by replacing the uniform weighting function on $g \in [0, 1]$ with a Jeffreys-style prior $\omega(g) \propto 1/g$ on $[\epsilon, 1]$. One can construct an orthonormal polynomial basis $\{\tilde{L}_m(g)\}$ with respect to this weight and repeat the derivation above. Multiplication by $g$ is again represented by a symmetric tridiagonal Jacobi matrix $\tilde{C}$, yielding dynamics of the same form as above.

However, the spectral consequences are subtler than a naive "prior over scales" interpretation might suggest. For any sufficiently regular positive weight on a compact interval, the empirical eigenvalue distribution of the associated Jacobi matrix converges, as $M \to \infty$, to a universal arcsine law on that interval. Specifically, Proposition 1.2 of (Kuijlaars & Van Assche, 1999) states that for a measure $\mu$ supported on a single interval $[\alpha, \beta]$ with positive density almost everywhere, the normalized zero counting measure of the degree-$M$ orthogonal polynomial converges weakly to the equilibrium arcsine measure on $[\alpha, \beta]$, with density

$$\frac{1}{\pi\sqrt{(\beta - g)(g - \alpha)}}\,\mathbf{1}_{(\alpha,\beta)}(g)\,dg.$$

Since the eigenvalues of the $M \times M$ Jacobi truncation coincide with the zeros of the degree-$M$ orthogonal polynomial, the empirical eigenvalue distribution of the Jacobi matrix converges to the same arcsine law. Consequently, reweighting the measure on $g$ while preserving compact single-interval support and positivity of the density does not change the limiting distribution of eigenvalues, and therefore does not fundamentally alter the characteristic timescales represented by the system. This motivates the alternative log-timescale construction described next.

**D.3. Log-Timescale Formulation ($u = \log g$)**

To obtain an embedding that allocates resolution uniformly across decades of timescale, we instead parameterize

$$u \triangleq \log g \in [\log \epsilon, 0],$$

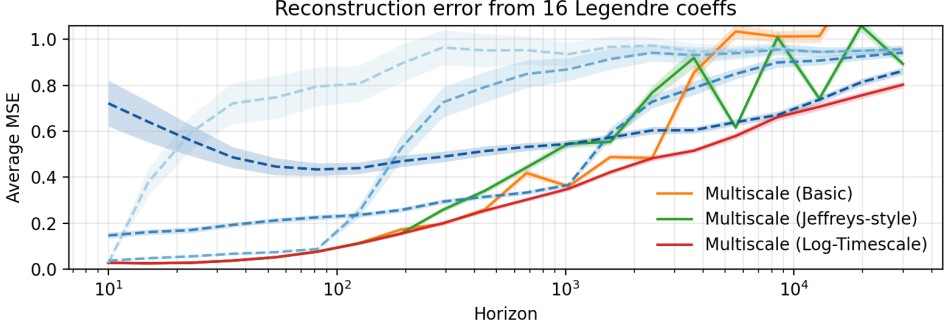

*Figure 7.* The log-timescale Multiscale HiPPO variant consistently outperforms the basic and Jeffreys-style variants on the variable-horizon reconstruction experiment from Section 3.4. Single-scale HiPPO systems with timescales $10^1, \dots, 10^4$ are shown in light to dark blue for comparison. Shaded regions denote $\pm$ SEM for 64 independently sampled trajectories of length $T = 3 \times 10^4$.

and represent the scale-dependent state as a function of $u$:

$$\mathbf{s}(t; u) \approx S(t)\, \boldsymbol{\psi}(u),$$

where $\{\psi_m(u)\}_{m=0}^{M-1}$ are orthonormal polynomials on $[\log \epsilon, 0]$ with respect to the uniform weighting.

The underlying HiPPO dynamics become

$$\dot{\mathbf{s}}(t; u) = e^u \big[ A\, \mathbf{s}(t; u) + \mathbf{b}\, f(t) \big].$$

In the $\{\psi_m\}$ basis, multiplication by $u$ is represented by a symmetric tridiagonal Jacobi matrix $J$, while multiplication by $g(u) = e^u$ is represented by the matrix function

$$G = \exp(J).$$

Substituting the expansion yields multiscale dynamics

$$\dot{S} = ASG + Bf(t),$$

with an analogous rank-1 input injection $B = \mathbf{b}\mathbf{r}^\top$, where $\mathbf{r}$ are the $\{\psi_m\}$-coefficients of $e^u$ (equivalently, $\mathbf{r} = G\mathbf{e}_0$). Unlike the basic construction, the matrix $G$ is generally dense, and the $\mathcal{O}(NM)$ update cost associated with a tridiagonal scale-coupling matrix is lost. Nevertheless, for moderate $M$, $G$ can be precomputed explicitly, or applied implicitly as a matrix function of $J$ using Krylov or polynomial approximation methods.

The advantage of the $u = \log g$ formulation is conceptual and practical: the spectrum of $J$ distributes over $[\log \epsilon, 0]$, and the spectrum of $G = \exp(J)$ is its exponential pushforward onto $[\epsilon, 1]$. This yields a representation whose effective timescales are approximately log-uniformly distributed, allowing substantially more resolution at long horizons for a fixed scale dimension $M$.

In the experiments reported in the main text, we use this log-timescale formulation, as it provides the most stable coverage of a wide range of temporal horizons under a fixed coefficient budget. See Figure 7 for an empirical comparison between the three multiscale systems.

### D.4. Relation to Banks of HiPPO Systems

Although the multiscale dynamics are written in a coupled form, they do not introduce fundamentally richer dynamics than a bank of independent HiPPO systems. In particular, the scale-coupling operator $G$ admits a spectral decomposition, under which the system decouples into a set of HiPPO systems operating at fixed effective timescales, up to a linear rotation of the input injection $B$. The advantage of Multiscale HiPPO is therefore not increased expressivity, but a structured and queryable parameterization that compresses a continuum of timescales into a single state with shared structure and controlled complexity.

## E. Forecast-Induced History Geometry and the Identity-Metric Case

This appendix justifies the coefficient-space approximation $Q \propto T^\top T$ used in the main text (Section 3.5) and in Figure 5. Throughout, let $h$ denote a history (the restriction of the signal to $(-\infty, t]$). Let $x(h) \in \mathbb{R}^n$ be a finite-dimensional encoding of $h$ (in our experiments, $x(h)$ is the HiPPO coefficient state of System 2/3), and let $y(h) \in \mathbb{R}^p$ be the target representation of the future over a horizon $H$ (in our experiments, $y(h)$ is the System 1 Legendre coefficient vector for the window $[t, t+H]$).

**General weighted forecasting losses and the induced bilinear form.** Consider a forecasting objective defined by a weighted squared error over the horizon:

$$\mathcal{L} = \mathbb{E}\left[\int_0^H w(\tau)\left(\hat{f}_h(\tau) - f(t+\tau)\right)^2 d\tau\right],$$

where $w(\tau) \geq 0$ is a weighting function. Suppose $\hat{f}_h(\tau)$ is represented in a basis $\{\phi_k(\tau)\}_{k=1}^p$ via a coefficient vector $\hat{y}(h) \in \mathbb{R}^p$:

$$\hat{f}_h(\tau) = \sum_{k=1}^p \hat{y}_k(h)\,\phi_k(\tau) \quad \text{and similarly} \quad f(t+\tau) \approx \sum_{k=1}^p y_k(h)\,\phi_k(\tau).$$

Then the weighted $L^2$ loss is a quadratic form in coefficient space,

$$\int_0^H w(\tau)\left(\hat{f}_h(\tau) - f(t+\tau)\right)^2 d\tau = \left(\hat{y}(h) - y(h)\right)^\top W\left(\hat{y}(h) - y(h)\right),$$

where the metric $W \in \mathbb{R}^{p \times p}$ is the weighted Gram matrix

$$W_{ij} \triangleq \int_0^H w(\tau)\,\phi_i(\tau)\,\phi_j(\tau)\,d\tau.$$

This immediately defines a positive semidefinite bilinear form on histories through predicted futures:

$$\langle h, h' \rangle \triangleq \mathbb{E}\left[\hat{y}(h)^\top W\,\hat{y}(h')\right].$$

**Linear predictors induce $Q = T^\top W T$.** In the linear Forecasting HiPPO setting, the predicted future coefficients are linear in the history state:

$$\hat{y}(h) = T\,x(h),$$

for some matrix $T \in \mathbb{R}^{p \times n}$. Substituting yields

$$\hat{y}(h)^\top W\,\hat{y}(h') = x(h)^\top \left(T^\top W T\right) x(h').$$

Thus the forecasting objective induces a quadratic form on history states,

$$Q \triangleq T^\top W T,$$

which describes how similarly two histories support forecasts under the chosen horizon and weighting.

**Why $W = I$ in our experiment (and hence $Q = T^\top T$).** In our experiments, the target $y(h)$ is the vector of Legendre HiPPO coefficients produced by System 1, and we train $T$ using a squared Euclidean loss in this coefficient space:

$$\mathbb{E}\left[\|y(h) - Tx(h)\|_2^2\right].$$

Because the System 1 coefficients correspond to an *orthonormal* Legendre basis on $[0, 1]$ under the uniform weighting function used by the HiPPO reconstruction, the natural coefficient-space metric is the identity:

$$W = I.$$

Therefore the induced history quadratic form simplifies to

$$Q = T^\top T.$$

(If one instead measured error in the time domain with a nonuniform $w(\tau)$, or used a non-orthonormal future representation, then $W$ would be nontrivial and the appropriate form would be $Q = T^\top W T$.)

**Connection to the estimators used in the experiment.** We estimate $T$ from streaming second-order statistics of the pair

$$(x, y)$$

($x = S_2(t)$ and $y = S_1(t)$). The full-rank least-squares map is

$$T_{\text{LS}} = \Sigma_{yx} \, \Sigma_{xx}^{-1},$$

and the rank-$d$ reduced-rank regression (RRR) map $T_d$ is obtained by truncating the singular directions of the whitened cross-covariance. Either choice yields a valid induced form $Q = T^\top T$ (full-rank or rank-$d$), and in Figure 5 we visualize this geometry by pushing the rank-$d$ $Q$ back to a lag-domain kernel via basis evaluation on a lag grid.

## F. Language Modeling Experiments

### F.1. Experimental Setup

We evaluate HiPPO Zoo variants on WikiText-2-raw (Merity et al., 2016), a standard character-level language modeling benchmark. The dataset contains 96 unique characters after preprocessing. We use the official train, validation, and test splits.

All models are parameter-matched at approximately 25k parameters (single-layer) or 40k parameters (two-layer baselines). We use standard minibatch training with the AdamW optimizer, a learning rate of $10^{-3}$ for all models, and early stopping on validation perplexity with patience set to five epochs. For SSM models (S4D, Mamba), we use default timescale parameters $\Delta t_{\min} = 0.001$, $\Delta t_{\max} = 0.1$.

### F.2. Results

Table 4 shows test perplexity for all models. HiPPO Zoo variants underperform modern SSMs, as expected. Vanilla HiPPO with MLP readout (6.80 PPL) approaches but does not match LSTM performance (5.46 PPL), while specialized variants (Salience, Associative Memory) show task-specific strengths on synthetic tasks but do not transfer to language modeling. Associative Memory HiPPO's catastrophic failure (20.41 PPL) demonstrates that continuous addressing is fundamentally mismatched to character-level autoregressive prediction of natural language.

*Table 4.* WikiText-2-raw character-level language modeling results.

| Model | Params | Test PPL |
|---|---|---|
| *Baselines (single-layer)* | | |
| LSTM | 25k | **5.46** |
| S4D | 25k | 5.75 |
| Mamba | 26k | 5.54 |
| *Baselines (two-layer)* | | |
| LSTM | 42k | 4.75 |
| S4D | 39k | 5.28 |
| Mamba | 43k | **4.65** |
| *HiPPO Zoo (single-layer)* | | |
| Vanilla HiPPO | 25k | 9.68 |
| Vanilla HiPPO + MLP | 25k | **6.80** |
| Salience HiPPO | 25k | 7.52 |
| Assoc. Memory HiPPO | 25k | 20.41 |

### F.3. Salience HiPPO vs. Mamba Timestep Analysis

To understand how scalar salience $g(t)$ relates to Mamba's vector-valued $\Delta t$ parameters, we analyzed their learned values on the WikiText-2 test set. Mamba produces 88-dimensional $\Delta t$ vectors per character (with `dt_rank = 6` in the implementation). We computed:

- PCA on Mamba's $\Delta t$: the first PC explains 79% of variance, suggesting approximate low-rank structure

- Linear regression predicting Salience $g(t)$ from Mamba $\Delta t$: $R^2 = 0.38$

- Linear regression predicting Mamba $\Delta t$ from Salience $g(t)$: $R^2 = 0.01$

The moderate correlation ($R^2 = 0.38$) suggests Salience and Mamba identify partially overlapping patterns of input importance. However, the asymmetry ($R^2 = 0.01$ for reverse prediction) indicates that Mamba's entrywise parameterization captures additional structure that cannot be reduced to a scalar.

We note that Salience HiPPO and Mamba differ in multiple ways beyond scalar vs. vector adaptivity: Mamba's $\Delta t$ is input-dependent only (for hardware efficiency), while Salience $g(t)$ is both input- and state-dependent. Isolating the effect of dimensionality would require ablation studies with scalar or state-dependent Mamba variants, which we leave to future work.

