# OpenReview forum: "HiPPO Zoo: Explicit Memory Mechanisms for Interpretable State Space Models"
_ICML.cc/2026/Conference — ICML 2026 regular_

### Official Review · Reviewer_YzGu · 2026-02-16

**Soundness:** 3
**Presentation:** 4
**Significance:** 3
**Originality:** 4
**Overall Recommendation:** 5
**Confidence:** 3

**Summary:**

The authors noticed that state space models (SSMs) in practice have evolved far beyond their original theoretical grounding of memory via online orthogonal polynomial projections (HiPPO).
To re-connect modern SSMs to HiPPO theory, the authors draw the following theoretical connections between the two, and confirm their findings in toy experiments:
1. The time-invariant nonlinear functions that deep SSMs pass state vectors through can be interpreted or reformulated via Volterra series.
2. The input-dependent recurrences used in selective SSMs (e.g., Mamba) can be interpreted as input-dependent warping of the HiPPO history measure, especially if inputs affect all state entries equally via a multiplicative scalar.
3. Content-dependent retrieval, a feature of selective SSMs, can be grounded in orthogonal polynomials.
4. The timescale equivariance of HiPPO enables creation of a multiscale HiPPO.
5. HiPPO can be formulated into an interpretable forecaster.

**Compliance With Llm Reviewing Policy:**

Affirmed.

**Final Justification:**

My primary concerns were about whether the authors' various HiPPO theories would hold in practical settings. While there is still some room for further evaluation of their ideas (for example, Volterra-HiPPO vs nonlinearly stacked SSM layers), the authors have done a good job evaluating their work in challenging settings like NLP. Their paper has potential to impact the field of RNNs.

**Key Questions For Authors:**

My sole question for the authors is: how well do your proposed ideas apply to deep SSMs trained on benchmarks like sequential image data or LRA tasks? In other words, what are the limitations of your proposed orthogonal polynomial perspectives on SSMs in practice?
For example:
* In a benchmark like sequential MNIST, where a ground-truth Volterra system is not known, how does a stack of Volterra HiPPO layers perform against a stack of HiPPO + MLP layers, in both performance and interpretability?
* In language tasks, where selective SSMs outperform linear SSMs, and where input-dependent recurrences are useful, how accurate is the salience signal $g(t)$ as an interpretable approximation of the entrywise multiplicative interactions common in architectures like Mamba? One would presume that replacing a multiplicative interaction at each entry of the SSM state with a single scalar multiplicative interaction should decrease performance, but are the losses in performance sufficiently small to justify the increase in interpretability gained by interpreting selective SSMs via history measure warping?
* How does an OP-based associative memory architecture compare to a selective SSM in a language task?
* What are some examples of practical scenarios or benchmarks in which multiscale SSMs are applicable, and how do multiscale SSMs perform in them?
* How does the proposed HiPPO forecaster compare to existing SSM forecasters on forecasting benchmarks?

**Limitations:**

The authors have not shown the limitations of their proposed ideas in practice (see "Key Questions for Authors"). For this reason, I would need more information about how the authors' proposed ideas work in SSMs trained on common benchmarks before I could rate the paper above 4.

**Strengths And Weaknesses:**

*Soundness:*
Each of the authors' ideas is straightforward, mathematically accessible (assuming familiarity with HiPPO theory), and confirmed via toy experiments. However, the degree to which the authors' proposed ideas can apply to SSMs in practice remains unclear.

*Presentation:*
The authors' work was clearly written. I appreciated that each section was neither too short nor too verbose, and each experiment was well-documented. The context of background in which this work exists was clear.

*Significance:*
Grounding modern SSMs in HiPPO theory is an open question that, if solved, should illuminate future avenues through which to improve nonlinear recurrent networks. The authors' findings are significant and promising, but their practical utility and limitations remain to be seen.

*Originality:*
In the context of modern SSMs, the authors' several proposed ideas are, to my knowledge, certainly novel, and hold potential for SSM research and practice.

*Weaknesses:*
The main weakness of the paper is a lack of validation of the authors' ideas on SSMs trained on datasets common in SSM evaluation (see "Key Questions for Authors").

---

> ### Author Rebuttal · Authors · 2026-03-31
>
> Thank you for your careful reading of our work. We are grateful for your recognition of the clarity of our presentation, the novelty of our ideas, and the potential significance of grounding modern SSMs in HiPPO theory. Your question about validating our ideas on common SSM benchmarks is both natural and important. While comprehensive evaluation on deep SSM architectures across multiple benchmarks is beyond the scope of this initial work, we agree that demonstrating practical applicability would strengthen the contribution. Below, we report experiments that begin to address your questions, focusing on settings where our explicit polynomial framework can be fairly compared to existing methods. We view deeper integration into modern SSM architectures as important future work and appreciate your suggestions for specific comparisons.
>
> ## Baseline comparisons on synthetic tasks
> We have added LSTM, Transformer, and S4D baselines to the selective copying and associative recall tasks. These experiments isolate specific memory mechanisms in controlled settings, allowing us to assess whether explicit polynomial structure provides advantages when task requirements align with the mechanism.
>
> **Selective Copying:** Salience HiPPO achieves 100% test accuracy, demonstrating that the scalar salience signal $g(t)$ successfully implements adaptive memory allocation despite being simpler than entry-wise gating in Mamba. Parameter-matched S4D achieves 81%, while 2-layer S4D reaches 97%. This suggests the explicit measure warping is competitive with and more interpretable than implicit pointwise nonlinearity-based selective mechanisms on this task.
>
> **Associative Recall:** Associative Memory HiPPO achieves 100% test accuracy, showing that explicit OP-based key-value memory can solve content-dependent retrieval. S4D achieves only 33% (both single- and 2-layer), suggesting this task requires mechanisms beyond standard linear SSMs with pointwise nonlinearities.
>
> Complete results table: anonymous.4open.science/r/hippo-zoo-C7C1/
>
> ## Language modeling on WikiText-2
> We are conducting experiments on WikiText-2-raw to evaluate HiPPO Zoo variants on a standard language modeling benchmark. This will provide concrete evidence addressing several of your questions:
> * How does scalar salience $g(t)$ compare to higher-rank selective mechanisms in Mamba on a practical task?
> * How does OP-based associative memory perform compared to modern SSMs in language modeling?
> * What is the practical performance gap between explicit HiPPO mechanisms and modern SSMs?
>
> We will report these results within the discussion period.
>
> ## Key Questions
> **1) Volterra HiPPO stacking on sMNIST:** We have not yet tested stacked Volterra layers on sequential MNIST. This is an excellent suggestion for demonstrating architectural scalability. We note that our synthetic Volterra experiment (Wray-Green system) demonstrates that quadratic Volterra HiPPO successfully recovers interpretable nonlinear kernels where linear models fail, but we agree that testing on standard benchmarks would strengthen this claim.
>
> **2) Scalar vs. vector salience:** The WikiText-2 experiments (in progress) will directly address this question. We expect some performance degradation from using scalar $g(t)$  versus full entry-wise gating, but the trade-off may be acceptable in settings where visualizing memory allocation matters. The selective copying results suggest the scalar approximation is sufficient when memory allocation patterns are uniform across state dimensions.
>
> **3) Associative memory vs. selective SSMs:** The WikiText-2 experiments will provide this comparison. We note that Associative Memory HiPPO uses explicit continuous addresses rather than implicit content-based retrieval, making the stored associations directly inspectable.
>
> **4) Multiscale applicability:** Multiscale HiPPO is most applicable in settings with unknown or variable temporal horizons, for example streaming applications where relevant timescales span orders of magnitude. Our variable-horizon reconstruction experiment demonstrates that a single Multiscale system achieves near-optimal reconstruction across 3+ orders of magnitude in timescale, while single-scale systems fail outside their design horizon.
>
> **5) Forecasting benchmarks:** We have not tested on standard forecasting benchmarks. Our Forecasting HiPPO contribution is primarily conceptual, showing how objectives induce interpretable geometry on histories, rather than proposing a competitive forecaster.
>
> ## Summary
> The synthetic task results demonstrate that explicit HiPPO mechanisms can achieve perfect accuracy when task-mechanism alignment is strong, even outperforming modern SSMs like S4D. The WikiText-2 experiments (in progress) will quantify performance trade-offs on real-world data. We view this work as establishing a foundation for interpretable SSM design, with integration into deep architectures and comprehensive benchmarking as natural next steps.

---

> > ### Author Rebuttal · Reviewer_YzGu · 2026-04-01
> >
> > I appreciate the authors' rebuttal, they acknowledged all my questions.
> >
> > The authors' new synthetic selective copy / associative recall experiments are compelling, and show the merit of their proposals. Nonetheless, I have a couple of questions:
> > 1. I assume transformers don't do well on associative recall and selective copy because of the parameter budget, is this correct?
> > 2. Why do none of the architectures aside from Associative Memory HiPPO do well on the associative memory task? Is this again related to the parameter budget?
> >
> > I look forward to seeing the results on WikiText-2 experiments, they will help address some important questions before I make a final decision on my rating.

---

> > > ### Author Response · Authors · 2026-04-04
> > >
> > > Thank you for these additional questions. First, please see our response to pSpo for the experimental results on WikiText-2-raw. Below, we discuss Salience HiPPO versus Mamba in more detail and then provide answers to your synthetic experiment questions.
> > >
> > > **Scalar salience vs. Mamba's vector timestep parameter:** We investigated whether scalar salience $g(t)$ in Salience HiPPO correlates with Mamba's learned entry-wise ``dt`` parameters to assess if they capture similar adaptive patterns on the WikiText-2 test set.
> > > * Mamba produced $d_\text{model} = 88$ input-dependent timestep parameters per character (with a ``dt_rank`` parameter of 6)
> > > * The first principal component ``dt`` explains 79% of its variance, suggesting an approximate low-rank structure
> > > * Predicting Salience $g(t)$ from Mamba ``dt`` via a linear predictor results in $R^2 = 0.38$ (moderate alignment)
> > > * Predicting Mamba ``dt`` from Salience $g(t)$ results in $R^2 = 0.01$ (low alignment)
> > >
> > > The moderate correlation ($R^2 = 0.38$) suggests Salience HiPPO and Mamba identify partially overlapping patterns of input importance, but the low correlation ($R^2 = 0.01$) suggests Mamba's entrywise parameterization captures additional structure. Of course, there are other differences in architectures of the two systems apart from the scalar vs. entrywise salience/``dt`` choices, which may result in the two signals playing overall less analogous roles in the two models. For example, Mamba's ``dt`` is solely input-dependent (for hardware acceleration), whereas Salience HiPPO's $g(t)$ is both input- and state-dependent. Additional investigations involving scalar or state-dependent valued ``dt`` variants of Mamba would be needed to more fully study the effect of these architectural differences.
> > >
> > > Our selective copying results suggest scalar salience works well when adaptive memory allocation is the primary requirement and patterns are relatively uniform across state dimensions. WikiText-2 performance (7.52 vs. 5.54 PPL for Mamba) suggests language modeling may benefit from the additional expressivity of entry-wise parameterization, though we note these architectures differ in multiple ways beyond scalar vs. vector adaptivity.
> > >
> > > ## Synthetic Task Follow-Up Results
> > >
> > > The following table investigates performance of different models on AR with more data, more capacity (model parameters), and more optimization steps. Additionally, transformers are tested with and without learned positional embeddings (PEs).The initial results were presented without positional embeddings.
> > >
> > > |Setting|Model|Test Acc|
> > > |---|---|---|
> > > |**HiPPO Zoo Baseline**|||
> > > |Baseline|Assoc Memory HiPPO|**100.0%**|
> > > |**Transformer Variants**|||
> > > |Baseline|Transformer|32.9%|
> > > |Baseline+PE|Transformer|33.2%|
> > > |Capacity 10x|Transformer|32.4%|
> > > |Capacity 10x+PE|Transformer-2|**98.9%**|
> > > |Data 10x|Transformer|32.9%|
> > > |Data 10x+PE|Transformer-2|32.7%|
> > > |Steps 2x|Transformer|34.5%|
> > > |Steps 2x+PE|Transformer-2|**99.7%**|
> > > |Capacity 10x+Data 10x+Steps 2x|Transformer|32.3%|
> > > |Capacity 10x+Data 10x+Steps 2x|Transformer-2|31.2%|
> > > |**LSTM and S4D**|||
> > > |Baseline|LSTM|32.7%|
> > > |Baseline|LSTM-2|32.0%|
> > > |Baseline|S4D|33.2%|
> > > |Baseline|S4D-2|33.2%|
> > > |Capacity 10x+Data 10x+Steps 2x|LSTM|**68.2%**|
> > > |Capacity 10x+Data 10x+Steps 2x|LSTM-2|**77.8%**|
> > > |Capacity 10x+Data 10x+Steps 2x|S4D|32.8%|
> > > |Capacity 10x+Data 10x+Steps 2x|S4D-2|32.8%|
> > >
> > > Below, we make several observations.
> > >
> > > **Transformers need architecture and scale**
> > > * Single-layer transformers cannot solve AR regardless of capacity/data/steps.
> > > * Two-layer transformers can solve AR (near-100% test accuracy) with more steps or more capacity and positional embeddings. This is intuitive given the task structure (find most recent occurrence of query token, select next token)
> > > *  Positional embeddings seem necessary to solve the task.
> > >
> > > **LSTM needs more scale**
> > > * LSTMs only achieve >60% accuracy with 10x capacity, 10x data, and 2x steps. Still, the accuracy is far from 100%, unlike high-performing transformer settings.
> > >
> > > **S4D is unable to solve AR**
> > > * Performance is stuck at < 35% regardless of capacity, data, and steps.
> > > * Linear recurrence with limited nonlinearities are unable to bind arbitrary token pairs.
> > >
> > > **Associative Memory HiPPO is efficient**
> > > * 100% accuracy with 25k parameters and baseline data
> > > * Demonstrates the value of task-aligned explicit structure

---

### Official Review · Reviewer_pSpo · 2026-03-07

**Soundness:** 1
**Presentation:** 3
**Significance:** 2
**Originality:** 3
**Overall Recommendation:** 3
**Confidence:** 2

**Summary:**

This paper extends the **HiPPO framework**, focusing on **multi-scale memory representations**, modeling of **nonlinear Volterra systems**, and **prediction-induced memory geometry**. The authors introduce **Multiscale HiPPO**, which represents historical information simultaneously across multiple time scales using continuous-time linear systems. This approach significantly improves upon the traditional single-scale HiPPO formulation.

In addition, the paper demonstrates the effectiveness of **Volterra HiPPO** for modeling **quadratic nonlinear systems**, and proposes **Forecasting HiPPO**, which shows how a prediction objective can induce a geometric structure in the representation of historical states. Overall, the work is technically solid, with detailed theoretical derivations and well-designed experiments. The proposed framework is both innovative and potentially useful for sequence modeling tasks.

**Compliance With Llm Reviewing Policy:**

Affirmed.

**Final Justification:**

While the theoretical synthesis is interesting and mathematically sound, the empirical execution does not yet meet the bar for publication. To be an impactful contribution, the paper either needs to narrow its focus to thoroughly validate 1-2 of these extensions, or significantly expand the experimental section to prove the efficacy and efficiency of the entire "zoo" against state-of-the-art baselines.

**Key Questions For Authors:**

see weakness

**Limitations:**

see weakness

**Strengths And Weaknesses:**

## Strengths

**1. Strong theoretical foundation and systematic framework**

The paper builds upon the HiPPO framework and systematically extends its representational power. It introduces several new directions, including multi-scale representations, nonlinear system modeling, and prediction-induced memory geometry. The theoretical derivations are clear and mathematically well-grounded.

**2. High level of novelty**

Multiscale HiPPO represents an important extension of the original HiPPO formulation, enabling the representation of multiple time scales simultaneously and addressing the limitations of single-scale systems in variable-scale tasks. In addition, **Volterra HiPPO** provides a new perspective for modeling nonlinear dynamical systems.

**3. Well-designed experiments**

The experimental evaluation includes tasks such as **quadratic Volterra system identification**, **selective copy**, **associative recall**, and **variable-scale reconstruction**. These experiments cover several key aspects of sequence modeling, ranging from nonlinear system identification to long-term memory representation. The results are generally convincing.

**4. Detailed appendix and strong reproducibility**

The appendix contains extensive technical details, including experimental setups and mathematical derivations. In particular, the derivations for **Multiscale HiPPO** and **prediction-induced geometry** are presented carefully, which significantly improves the reproducibility and clarity of the work.


## Weaknesses

My main concern with the paper is the **overall coherence and focus of the contributions**. The work introduces several extensions to the HiPPO framework (e.g., Multiscale HiPPO, Volterra HiPPO, and Forecasting HiPPO), but each individual modification appears relatively incremental and limited in scope. While each idea is interesting on its own, the paper combines many different directions without a clear unifying focus. As a result, the contributions feel somewhat fragmented rather than forming a single coherent research narrative. I am not sure if a zoo can be regarded as an academic article.

In addition, the experimental validation for each proposed component is relatively limited. Since the paper attempts to introduce multiple extensions simultaneously, none of them are explored in sufficient depth experimentally. Strengthening the empirical evaluation for each component, or focusing on a smaller number of contributions with more thorough validation, would make the work significantly stronger.

**1. Lack of baseline comparisons in some experiments**

For tasks such as **selective copy** and **associative recall**, the paper does not include comparisons with widely used sequence models such as **LSTM, Transformer, or S4**. Without these baselines, it is difficult to evaluate the relative advantages of the proposed methods.

**2. Limited discussion of computational efficiency**

Although the paper mentions that Multiscale HiPPO can effectively “compress multiple time scales,” it does not provide a detailed analysis of **computational cost**, **training stability**, or **scalability** to larger tasks.

**3. Intuition behind prediction-induced geometry could be clearer**

While the mathematical derivations are rigorous, the intuition behind how **prediction objectives induce geometric structure in the historical state representation** may be difficult for readers to grasp. Additional explanations, visualizations, or intuitive examples would make this concept easier to understand.

**4. Limited discussion of real-world applications**

The paper is largely theoretical and does not explore how the proposed models perform on practical tasks such as **language modeling** or **real-world time series prediction**. Including such experiments, or at least discussing potential applications, would strengthen the practical impact of the work.

---

> ### Author Rebuttal · Authors · 2026-03-31
>
> We appreciate your positive appraisal of the novelty, quality of experiments, theoretical foundation, and potential utility of our work. We respectfully disagree that a systematic framework cannot be a contribution. Our work asks: which modern SSM capabilities can be made interpretable within HiPPO? Each extension answers this question for a different capability (nonlinearity via Volterra, selectivity via Salience, content-addressability via Associative Memory, multiscale coverage via Multiscale, and objective-dependent memory via Forecasting). The unifying thread is explicit polynomial memory. All extensions preserve direct interpretability through the OP basis. That said, we acknowledge the evaluation could be stronger and address your concerns below.
>
> ## Lack of baselines
> We have added LSTM, Transformer, and S4D baselines for the selective copying and associative recall tasks. Results show that when task requirements align with explicit mechanisms, specialized HiPPO variants achieve perfect accuracy: Salience HiPPO reaches 100% on selective copying, while Associative Memory HiPPO reaches 100% on associative recall. Parameter-matched baselines do not reach perfect performance on either task (S4D achieves 81% and 33% respectively), and even 2-layer variants do not solve associative recall (33%), suggesting these tasks probe genuinely distinct memory mechanisms. A complete results table is in the repository README: anonymous.4open.science/r/hippo-zoo-C7C1/
>
> ## Limited discussion of computational efficiency
> You are correct that we should discuss computational costs more explicitly. We will highlight the following key points. 1) Multiscale HiPPO with log-timescale formulation requires $\mathcal{O}(NM^2)$ per update versus $\mathcal{O}(NM)$ for tridiagonal variants, trading efficiency for explicit scale coverage. 2) Volterra HiPPO has $\mathcal{O}(N^k)$ parameters for order-$k$ systems, limiting scalability to higher orders without low-rank or other structured approximations. 3) Salience and Associative Memory HiPPO add modest overhead (salience network parameters, associative coefficient storage) compared to vanilla HiPPO. We will add this analysis to the main text.
>
> ## Prediction-induced geometry intuition
> Thank you for this feedback. We agree the geometric intuition could be clearer. The key idea is: different forecasting objectives prioritize different aspects of history. A short-horizon forecaster needs fine temporal detail at recent lags (to predict the immediate next step), while a long-horizon forecaster needs smoother, longer-range structure. The matrix $Q = T^\top T$ captures this intuition mathematically. Its eigenfunctions reveal which directions in history are important for prediction purposes. We have added a schematic diagram (see repo) illustrating the three-system architecture and will expand the intuitive explanation in the main text.
>
> ## Discussion of real-world applications
> We are conducting experiments on WikiText-2-raw, a standard language modeling benchmark, to demonstrate performance on real-world data. This will provide concrete evidence of how HiPPO Zoo variants perform on practical tasks compared to standard baselines. We view this as an important step toward understanding the interpretability-performance trade-offs in realistic settings and will report results within the discussion period.
>
> ## Coherence and Depth
> Regarding your concern about coherence: we view the "zoo" framing as a strength, not a weakness. Each extension demonstrates that a specific modern SSM capability (nonlinearity, selectivity, associative recall, multiscale coverage, predictive memory) can be made explicit and interpretable. The paper's contribution is showing these capabilities are not inherently opaque—they can be realized within a principled polynomial framework. This systematic decomposition provides a toolkit for interpretable sequence modeling.
>
> Regarding experimental depth: we agree that comprehensive evaluation of each variant would strengthen the work. Our baseline comparisons and (ongoing) WikiText-2 experiments address this concern by demonstrating both task-specific effectiveness (synthetic tasks) and real-world applicability (language modeling). We view deeper investigation of each variant, including theoretical analysis, scaling studies, and integration into larger architectures, as important future work that builds on this foundation.

---

> > ### Author Rebuttal · Reviewer_pSpo · 2026-04-03
> >
> > I thank the authors for their response and for articulating the unifying thread of their work—namely, exploring which modern SSM capabilities can be made interpretable via explicit polynomial memory. I appreciate the conceptual ambition of building a systematic framework around HiPPO. However, after carefully considering the rebuttal, my core concerns regarding the depth of empirical validation and the paper's fragmented execution remain unresolved. Therefore, I will maintain my original score.
> >
> > Here are the specific reasons why my assessment remains unchanged:
> >
> > * **Breadth at the Expense of Depth:** While I acknowledge that "explicit polynomial memory" conceptually connects these modifications, introducing five distinct extensions (Volterra, Salience, Associative Memory, Multiscale, Forecasting) in a single paper inevitably forces a severe trade-off between breadth and depth. A paper proposing a "systematic framework" requires systematic and rigorous empirical validation for *each* proposed component. Currently, the ideas are introduced rapidly but left without the thorough stress-testing required to convince the community of their individual or collective efficacy.
> > * **Absence of Critical Baselines:** The authors acknowledge that the evaluation could be stronger, but the core issue is that claims of matching "modern SSM capabilities" are not backed by direct comparisons. Tasks like selective copy and associative recall absolutely require benchmarking against strong, standard baselines—such as Transformers, LSTMs, and crucially, modern SSMs like S4 or Mamba. Without these comparisons, it is impossible to gauge whether this interpretable framework is actually competitive or practically useful.
> > * **Missing Efficiency and Real-World Validation:** The current landscape of sequence modeling is heavily driven by hardware efficiency and scalability to massive real-world datasets (e.g., language modeling, long-horizon time series). Proposing a suite of new HiPPO variants without a concrete analysis of their computational costs, training stability, or performance on standard real-world benchmarks leaves the practical impact of this framework purely theoretical.
> >
> > While the theoretical synthesis is interesting and mathematically sound, the empirical execution does not yet meet the bar for publication. To be an impactful contribution, the paper either needs to narrow its focus to thoroughly validate 1-2 of these extensions, or significantly expand the experimental section to prove the efficacy and efficiency of the entire "zoo" against state-of-the-art baselines.

---

> > > ### Author Response · Authors · 2026-04-04
> > >
> > > Thank you for your response. Below, we present results on some real-world data (WikiText-2-raw).
> > >
> > > ## WikiText-2-raw results
> > >
> > > We ran experiments on the WikiText-2-raw dataset using the official train, validation, and test splits. We chose the "raw" variant with a standard character set, resulting in a dataset with 96 unique characters. The baseline and HiPPO Zoo models are parameter matched at 25k parameters and we additionally test more standard two-layer variants with 40k parameters for the baseline models. Learning rates were fixed at 1e-3 for all models, and early stopping on validation perplexity with a patience of 5 epochs was implemented.
> > >
> > > |Model|#Params|Test PPL|
> > > |---|---|---|
> > > |**Baselines (single-layer)**|||
> > > |LSTM|25,341|**5.46**|
> > > |S4D|25,119|5.75|
> > > |Mamba|25,572|5.54|
> > > |**Baselines (two-layer)**|||
> > > |LSTM-2|41,901|4.75|
> > > |S4D-2|39,141|5.28|
> > > |Mamba-2|42,556|**4.65**|
> > > |**HiPPO Zoo (single-layer)**|||
> > > |Vanilla HiPPO|25,440|9.68|
> > > |Vanilla HiPPO+MLP|25,429|**6.80**|
> > > |Salience HiPPO|25,337|7.52|
> > > |Assoc Memory HiPPO|25,438|20.41 (failed)|
> > >
> > > As expected, HiPPO Zoo variants underperform modern SSMs on WikiText-2. Vanilla HiPPO with MLP readout achieves 6.80 test perplexity versus 5.54 for Mamba (23% gap), while Salience HiPPO achieves 7.52 (36% gap). These results quantify the interpretability-performance trade-off: explicit polynomial memory enables direct analysis of what is remembered, but at a cost in raw performance. Notably, Associative Memory HiPPO fails catastrophically (20.41 PPL), suggesting its continuous addressing strategy is poorly suited to autoregressive prediction. These findings validate our positioning: HiPPO Zoo provides interpretable alternatives for settings where understanding memory mechanisms matters, not competitive replacements for state-of-the-art SSMs.

---

### Official Review · Reviewer_8TcV · 2026-03-08

**Soundness:** 3
**Presentation:** 4
**Significance:** 2
**Originality:** 4
**Overall Recommendation:** 5
**Confidence:** 4

**Summary:**

This paper introduces the "HiPPO ZOO" a collection of five extensions to the High-order Polynomial Projection Operators (HiPPO) framework. The primary objective is to bridge the gap between the interpretable but restricted linear memory of original HiPPO and the expressive but opaque mechanisms of modern State Space Models (SSMs) like Mamba. By recastng contemporary SSM capabilities—such as nonlinearity, selectivity, and multiscale processing—as explicit modifications to the underlying measure or dynamics of orthogonal polynomial (OP) bases, the authors provide a principled lens for mechanistic interpretability.

**Compliance With Llm Reviewing Policy:**

Affirmed.

**Final Justification:**

The author's addressed all my doubts and questions. The paper was quite strong to begin with. I happily raised my score from Weak Accept to Accept.

**Key Questions For Authors:**

1. How does the latency of the Multiscale HiPPO's dense $G$ matrix scale as the number of timescales $M$ increases, and does this negate the benefits of a single-state representation?
2. Given the $R^{N^k}$ parameter growth, have you explored low-rank approximations for the $k$-th order tensors to make Volterra HiPPO viable for higher orders?
3. Why were no experiments performed on standard benchmarks (e.g., Long Range Arena or WikiText) to quantify the exact "performance gap" between these explicit models and modern SSMs?
4. In the Associative Memory HiPPO, how do you manage memory interference when the number of stored key-value pairs exceeds the number of coefficients ($n_{assoc}$)?

**Limitations:**

The authors adequately identify that their models are for low-memory and interpretable regimes rather than high-performance benchmarks. However, they should more explicitly discuss the efficiency loss in the log-timescale variant and the scaling limitations of the Volterra series.

**Strengths And Weaknesses:**

Soundness

Strength: The mathematical foundation—rooted in Favard’s theorem and the properties of Jacobi matrices—is rigorous and provides a clear path for discrete-time implementation via zero-order hold (ZOH).
Weakness: There are notable computational efficiency trade-offs. For instance, the log-timescale Multiscale HiPPO requires a dense matrix $G$, which causes the system to lose the $O(NM)$ update efficiency characteristic of tridiagonal scale-coupling variants.

Presentation

Strength: The paper excels at visualizing abstract concepts, such as "history measure deformation" and "predictive history metrics".
Weakness: The reliance on synthetic tasks to validate the "zoo" makes it difficult to assess how these explicit mechanisms would behave under the noise and complexity of high-dimensional, real-world data.

Significance

Strength: It offers a principled framework for mechanistic interpretability, which is a growing concern in the sequence modeling community.
Weakness: The authors admit that modern SSMs can outperform these lightweight HiPPO systems on standard benchmarks. This limits the current significance to a "complementary regime" focused on scientific and streaming settings rather than competitive state-of-the-art predictive performance. Additionally, the Volterra extension faces a parameter explosion ($R^{N^k}$) for higher-order interactions, potentially limiting its practical scalability.

Originality

Strength: Recasting disparate behaviors like time-warping and associative recall into a unified OP framework is highly original.
Weakness: While original in its interpretation, the work primarily revisits and extends the existing HiPPO framework rather than introducing a fundamentally new architecture intended to replace current leaders like Mamba.

---

> ### Author Rebuttal · Authors · 2026-03-31
>
> Thank you for your thoughtful and technically detailed review. We appreciate your recognition of the mathematical rigor, quality of visualizations, and originality of capturing modern SSM behaviors in a unified and explicit OP framework. We agree with your assessment that quantifying the performance gap on standard benchmarks would strengthen the practical significance of this work. Below, we address your specific technical questions about computational efficiency, parameter scaling, and memory interference.
>
> ## Competitive Performance
> We have added baseline comparisons on the selective copying and associative recall tasks, including LSTM, Transformer, and S4D models parameter-matched to HiPPO variants (~25K parameters). Results show that when task requirements align with explicit mechanisms, specialized HiPPO variants can match or exceed general-purpose models: Salience HiPPO achieves 100% accuracy on selective copying, while Associative Memory HiPPO achieves 100% on associative recall. Parameter-matched S4D reaches 81% and 33% respectively, while deeper 2-layer variants reach 97% and 33%, suggesting these tasks genuinely probe distinct memory mechanisms.
> We are also conducting experiments on WikiText-2 to provide concrete performance numbers on a standard language modeling benchmark. This will quantify the interpretability-performance trade-offs you identify between explicit HiPPO variants and modern SSMs. We will report these results within the discussion period.
>
> ## Key Questions
> **1. Multiscale HiPPO efficiency:** The dense-$G$ log-timescale variant costs $\mathcal{O}(NM^2)$ per update versus $\mathcal{O}(NM)$ for the tridiagonal basic variant. For the $M=128$ scale dimension in our experiment, precomputing $G=\exp(J)$ is practical. For very large $M$, Krylov methods could approximate the matrix exponential efficiently. We view this cost as appropriate for applications requiring explicit scale coverage across multiple orders of magnitude, but acknowledge the efficiency trade-off. We will discuss this more explicitly in the limitations.
>
> **2) Low-rank Volterra kernel approximations:** We have not yet explored low-rank parameterizations for the Volterra kernel tensor, although it is a natural direction for mitigating the $\mathcal{O}(N^k)$ parameter growth. For instance, Tucker or CP decompositions could parameterize $\beta^{(k)}$ with $\mathcal{O}(kNr)$ parameters for rank $r$, enabling higher-order interactions at modest cost. We will discuss this as future work and add it to the limitations section.
>
> **3 Standard benchmark experiments:** As noted above, we are conducting WikiText-2 experiments to quantify performance gaps. Our initial focus on synthetic tasks was intentional—they isolate specific memory mechanisms for direct analysis, but we agree that real-world validation strengthens the contribution.
>
> **4) Associative memory interference:** When the number of stored key-value pairs exceeds $n_\text{assoc}$, interference is governed by the OP reproducing kernel $K(x,x') = \phi(t)^\top \phi(t')$. Values stored at nearby addresses interfere according to this kernel, which decays with address distance but has nonzero overlap. In our experiments ($n_\text{assoc} = 32$, ~6 associations stored), the system learns to space addresses appropriately to minimize interference. For settings requiring more associations, one could increase $n_\text{assoc}$ or implement a strategy where older associations are selectively overwritten based on recency or importance scores.
>
> ## Additional Updates
> We have also added: (1) quantitative results tables with test accuracies for all methods (see repo: anonymous.4open.science/r/hippo-zoo-C7C1/), (2) updated Multiscale HiPPO figure with oracle error bounds showing near-optimal performance, and (3) Volterra experiment with error bars across multiple data realizations. These additions address the evaluation concerns you identified.

---

> > ### Author Rebuttal · Reviewer_8TcV · 2026-04-01
> >
> > I would like to thank the authors for their detailed and technically robust rebuttal. The addition of parameter-matched baseline comparisons (LSTM, Transformer, and S4D) on selective copying and associative recall tasks significantly strengthens the paper’s claims, demonstrating that these explicit HiPPO mechanisms can indeed outperform general-purpose models when task requirements align with the underlying memory dynamics.
> >
> > The clarification regarding the $\mathcal{O}(K^2)$ cost of the Multiscale variant and the potential for low-rank Volterra approximations provides a much clearer picture of the scalability trade-offs. Furthermore, the commitment to include WikiText-2 results and the additional quantitative tables in the final manuscript addresses my primary concern regarding the "performance gap" between interpretable and state-of-the-art models.
> >
> > Given that these new experiments and clarifications will be incorporated into the final version, I am pleased to increase my score. The "HiPPO ZOO" provides a highly original and mathematically rigorous contribution to the mechanistic interpretability of state space models.
> >
> > I am raising my score to 5 (Accept)

---

### Official Review · Reviewer_YevX · 2026-03-15

**Soundness:** 3
**Presentation:** 2
**Significance:** 3
**Originality:** 3
**Overall Recommendation:** 4
**Confidence:** 4

**Summary:**

This paper extends the HiPPO framework with 5 extensions:
- the Volterra HiPPO, which uses the HiPPO state vector in Volterra kernels truncated at some degree as the readout;
- Salience HiPPO, which can be interpreted as reparameterising time;
- Associative memory HiPPO, which use different learned projections to encode temporal encoding and write address;
- Multiscale HiPPO, which represents the HiPPO state as a function of an inverse timescale parameter (on log scale);
- Forecasting HiPPO, which maintains 3 separate HiPPO systems to exact different time based information.

For each of these extensions, there is an associated experiment that shows off its advantage.

**Compliance With Llm Reviewing Policy:**

Affirmed.

**Final Justification:**

In the rebuttals, the authors have adequately addressed my concerns relating to soundness, including releasing the code, showing errors/robustness to seeds. I see that also productive dialogue with other reviewers, and the authors have additionally evaluated these architectures for other problems. Overall, with these changes, this paper would make a good contribution.

The reason for not a higher score is that from the final response from the authors, a slight reservation is that the original code did not fix the numpy seed so some existing results that we see in the draft may be subject to change.

**Key Questions For Authors:**

For the equation of the Volterra HiPPO readout, there are multiple HiPPO state vectors multiplied together. What exactly are the dimensions of $s_n$ and what does multiplication of these vectors mean? Where do the weights go?

Similarly, for the prediction set-up, what is the dimension of $f_h$ and what does it mean to multiply them together in the bilinear form?

For Forecasting HiPPO, it is mentioned that "short-term forecaster emphasizes fine detail at small lags, while the long-horizon forecaster retains smoother structure extending further into the past ... also reflect in the leading eigenfunctions". How are these reflected in Figure 5? It seemed to me that predictive HiPPO memory for horizon was less accurate over longer time, e.g. at relative time $-30$.

Is there a trade-off between these specialised HiPPOs? Are they less general? If there is a situation where two of more variations make sense, how should one decide?

**Limitations:**

Limitations are not discussed. But related to my final question above, it would be nice to discuss the limitations of each type of HiPPO, would this mean they are each less "general" and how to choose between these.

**Strengths And Weaknesses:**

__Soundnesss__

One key issue is that the experimental details are thin, for a lot of the experiments, only a single example is provided (see Figure 2 and 3).

Another key issue is that code is not included as part of the supplementary materials or an anonymous link. The lack of code makes it impossible to reproduce the results, difficult to check the set-up to understand the experiments better, or determine if the examples we are seeing are "typical".

Without full experimental details or code, there are many claims that the readers are expected to believe. For example, we don't see anything indicating, e.g. proportion of correct retrieval for associative memory, simply told that they are able to "match the correct associated tokens". Is this suppose to be 100% accurate?

There is no description of Figure 1B (left). I presumed this is cumulative loss over training steps. It seems strange that 3 different set-ups would have exactly the same cumulative error at the start, is this a peculiarity of the seed used, or is there some reason you might expect this to happen? Repeating the experiments multiple times and representing the confidence interval on the plot would give greater confidence to the soundness of this plot.

A claim that I cannot verify is "associative memory HiPPO solves this tasks with a modest number of associative coefficient". The appendix suggest this was $32$, but without things like number of parameters or runtime of the model, it is hard to understand if this is indeed a modest number.

Another claim is that "Multiscale HiPPO produces stable reconstructions across all queried scales", however in Figure 4, we see that the average MSE clearly deteriorates for a longer horizon. The next sentence about improvement over the baselines is true, but it is unclear what "stable reconstructions" mean in this context.

Some extensions seem simplistic. The salience HiPPO introduces reweighting function $g(t)$, and multiplies this with $\dot{s}(t)$. Key questions include how to learn the best reweighting function and how stable this is. These are not addressed.

__Originality__

Creating and collecting these different types of HiPPO set-up seem new and interesting.

The experimental results, although often do not seem very surprising, does add nice narrative. For the Volterra HiPPO, it is shown that to capture a second order Volterra system, a first order system is obviously not enough, and that the second order Volterra HiPPO allows interpretation by recovering the kernel.

__Significance__

If the soundness issues can be fixed, I think introducing such a zoo of HiPPO variant is an interesting idea that others can build on (e.g. creating new variants, more analysis and theoretical properties of these) and would have some impact.

__Presentation__

There are a lot of assertions without references in the introduction, e.g. "contemporary SSMs routinely employ input-dependent state updates, adaptive allocation of memory across timescales, nonlinear interactions ..." but there is not a single reference to any of these SSMs.

It is not always clear how/what is being learned. E.g. for salience HiPPO, we may guess that $g(t)$ will be learned somehow, but it is not until the experiment details that the set-up of learning it is discussed. That discussion should move to an earlier part.

There are some important parts missing from some figures, Figure 4 (bottom) does not have a legend to explain the different lines.

Minor:
$w(x)$ is written first as "weighting function or probability density" but later is called a "measure". If it needs to be a measure, then be clear at its introduction.

Remove spurious "s" page in "vector s provides"

---

> ### Author Rebuttal · Authors · 2026-03-31
>
> Thank you for your detailed review. We appreciate your recognition that the manuscript presents new and interesting ideas with a nice narrative, and that these could be built upon in future work. We address your questions and concerns below.
>
> ## Code
> We apologize for not including code in the original submission. We have uploaded anonymized code reproducing all experimental results at: anonymous.4open.science/r/hippo-zoo-C7C1/
>
> ## Quantitative Results
> You are absolutely right. We have added quantitative metrics for HiPPO methods and parameter-matched baselines on the associative recall (AR) and selective copying (SC) tasks (see repo). Associative Memory HiPPO and Salience HiPPO achieve 100% test accuracy on AR and SC respectively, with 25k parameters (0.1 MB at float32). The examples shown are representative of typical learned behavior.
>
> ## Figure 1B Explanation
> You are correct this shows cumulative loss over training steps. The three methods start identically because they begin with zero-valued states and identical parameter initializations for the linear and quadratic Volterra HiPPO models. We have re-run this experiment with error bars (mean +/- SEM) across 5 random data realizations and parameter initializations (see repo) and will clarify the caption.
>
> ## "Stable Reconstructions"
> You are correct that "stable" was imprecise. We mean Multiscale HiPPO maintains consistent relative performance across scales, unlike single-scale systems that fail outside their design horizon. We will rephrase this. The subfigure now includes a legend and a lower bound on achievable MSE, showing Multiscale HiPPO achieves near-optimal reconstruction errors across scales (see repo).
>
> ## "Simplistic" Salience
> We respectfully view simplicity as a strength for interpretability. The scalar salience is the minimal modification enabling dynamic memory allocation while remaining interpretable. We will clarify the learning procedure earlier in the main text. The salience signal $g(t)$ is produced by an MLP taking the current input and pooled memory summary as input (Appendix C.2). This network converges stably via AdamW with learning rate $10^{-3}$.
>
> ## Presentation
> Thank you for detailed presentation feedback: adding citations, adding a legend to Figure 4 (bottom), using "weighting function" consistently for $w(x)$, and fixing typos. We will correct each.
>
> ## Key Questions
> **1) Volterra dimensions:** For Volterra HiPPO, $\mathbf{s}(t) = [s_0(t), \dots, s_{N-1}(t)]^\top$ is a column vector of size $N$, so each $s_n(t)$ is a scalar-valued function of time $t$. We apologize that the Volterra readout equation mistakenly used $\mathbf{s}_{i_1}(t)$, incorrectly suggesting each $s_n(t)$ is vector-valued. We will fix this in the text.
>
> **2) Forecasting bilinear form:** In Forecasting HiPPO, both $f(t)$ (true signal) and $\hat{f}_h(\tau)$ (predicted future) are scalar functions. The history $h$ is a restriction of $f$ to $(-\infty, t]$. We will make the time-dependence of $h$ explicit.
>
> **3) Forecasting eigenfunctions:** Thank you for noting this confusion. While the short-horizon predictive memory is more accurate than long-horizon at relative time -30, we intended to highlight: short-horizon remains fairly accurate (error < 0.2) until relative time −7, while long-horizon remains accurate until −13. These differences appear in the leading eigenfunctions of the two $Q$'s (right), where energy spreads over different time intervals. We will clarify the relevant time ranges.
>
> **4) Tradeoffs between variants:** This is an excellent question. Each variant makes a specific capability explicit at the cost of specialization. Volterra HiPPO exposes nonlinear dependence but has $\mathcal{O}(N^k)$ parameter growth for order-$k$ systems -- use when interpretable kernels are valued over black-box MLPs. Multiscale HiPPO's log-timescale variant requires a dense matrix $G$, trading $\mathcal{O}(NM)$ tridiagonal efficiency for $\mathcal{O}(NM^2)$ -- use when supporting unknown or variable timescales explicitly matters more than single-scale or bank-of-timescales efficiency.
>
> If predictive performance is paramount, use modern SSMs like Mamba or S4D. If understanding specific mechanisms is paramount, use corresponding HiPPO variants. If both are needed, consider hybrid architectures (future work). Variants can be combined: e.g., Multiscale applied to Forecasting produces scale- and horizon-dependent predictive memories. We view the zoo as a toolkit where practitioners choose interpretability/performance trade-offs appropriate for their application. We will add a discussion of these trade-offs to the conclusion.

---

> > ### Author Rebuttal · Reviewer_YevX · 2026-04-04
> >
> > Thank you for providing the code and clearer results in the repo. I have some remaining questions.
> >
> > Can you explain how the hyperparameters are selected for each set-up?
> >
> > I noticed also that in the updated Volterra figure, the shape at the start as well as where the MLP converge to seem to have changed quite a bit after being averaged over 5 seeds. E.g. the previous value for MLP in the draft does not even appear to be within 2 standard deviations away. Was the previous seed also included in these 5 seeds?
> >
> > I don't believe the authors quite addressed my initial point (perhaps tangentially through the trade-off between the variants in terms of number of params): _A claim that I cannot verify is "associative memory HiPPO solves this tasks with a modest number of associative coefficient". The appendix suggest this was , but without things like number of parameters or runtime of the model, it is hard to understand if this is indeed a modest number._

---

> > > ### Author Response · Authors · 2026-04-05
> > >
> > > Thank you for your continued engagement and for reviewing the updated materials. We address your questions below.
> > >
> > > **Hyperparameter Selection:** For the original submission experiments, we used reasonable default hyperparameters without extensive tuning. Learning rates were selected from {3e-4, 1e-3, 3e-3, 1e-2, 3e-2} based on preliminary runs. HiPPO timescale parameters are matched to episode length or, for Associative Memory HiPPO, a smaller context sufficient for reading and writing to the associative memory based on the recent past (``base_timescale = 2``). Network architectures use simple MLPs with hidden sizes in {128, 256}. We used reasonable defaults to demonstrate that HiPPO variants achieve strong performance without extensive tuning when task-mechanism alignment is good.
> > >
> > > For the baseline comparison experiments (added in the rebuttal), we performed more systematic model selection to ensure fair comparisons. All models were parameter-matched to ~25K parameters by adjusting hidden dimensions. Learning rates were selected from {3e-4, 1e-3, 3e-3} via validation loss, choosing the configuration with lowest validation MSE after 4,000 training steps. All other hyperparameters (optimizer = AdamW, weight decay = 1e-4, batch size) used standard defaults across all methods. This protocol ensures that performance differences reflect architectural capabilities rather than hyperparameter tuning advantages.
> > >
> > > **Figure 1B:** Thank you for pointing this out. The original figure showed a single run. The updated figure averages over 5 runs with different random seeds controlling both data generation and parameter initialization. One run reuses the old MLP initialization seed, but the original code did not fix the NumPy seed for input data generation, so that run uses different input data. The original single-run trace falls outside the error bars because it showed fast MLP convergence and low errors at the beginning of training. We note the shaded region represents +/-1 SEM, not standard deviation.
> > >
> > > **"Modest number of associative coefficients"**
> > > Here is comparative evidence for our "modest" number of associative coefficients claim:
> > >
> > > |Model|Parameters|AR Test Accuracy| State Dim |
> > > |---|---|---|---|
> > > |Assoc Memory HiPPO| 25,186 | 100.0% | 32 assoc + 32 HiPPO = 64|
> > > |LSTM (single-layer)| 24,620 | 32.7% | 52 |
> > > |LSTM (10x capacity, 10x data, 2x steps) | 246,500 | 68.2% | 172 |
> > > |Transformer-2 + PE | 51,196 | 99.7% | no persistent state |
> > > | S4D (any configuration) | 24,849-481,454 | ~33% | 75-310 |
> > >
> > > We mean "modest" in absolute terms (128 bytes), relative to the task (storing ~6 associations), and relative to other models (achieves 100% accuracy while LSTM needs 10x capacity and > 2.5x state coefficients to reach 68%).

---

### Decision · Program_Chairs · 2026-04-30

**Decision:**

Accept (regular)

**Comment:**

This paper presents a "HiPPO zoo" that makes several memory mechanisms from modern state space models explicit within an orthogonal-polynomial framework. Reviewers generally agreed that the paper is original, mathematically solid, and interesting from an interpretability perspective. The main concerns were limited empirical depth, missing baselines, the lack of code, limited discussion of computational tradeoffs, and uncertainty about how well the ideas extend beyond synthetic tasks.

During the discussion, the authors substantially strengthened the paper by releasing anonymous code, adding quantitative results and parameter-matched baselines on selective copying and associative recall, clarifying the seed issue, expanding the discussion of tradeoffs and limitations, and reporting WikiText-2 results that better position the work relative to modern SSM baselines. These additions addressed the concerns for most reviewers, although one reviewer still felt that the empirical support was not strong enough.

Overall, I think this is a solid and theoretically grounded paper with a clear interpretability focus, and I recommend acceptance. The final version should incorporate the main clarifications and added results from the rebuttal, and should state more explicitly that the goal of the paper is to make memory mechanisms explicit and analyzable rather than to compete with modern SSMs on standard benchmarks.